# Correlation between musculoskeletal structure of the hand and primate locomotion: Morphometric and mechanical analysis in prehension using the cross- and triple-ratios

Toshihiro Tamagawa[1], Torbjörn Lundh[2], Kenji Shigetoshi[3], Norihisa Nitta[4], Noritoshi Ushio[4], Toshiro Inubushi[5], Akihiko Shiino[6], Anders Karlsson[2], Takayuki Inoue[7], Yutaka Mera[8], Kodai Hino[1], Masaru Komori[9], Shigehiro Morikawa[6], Shuji Sawajiri[1], Shigeyuki Naka[10], Satoru Honma[11], Tomoko Kimura[1], Yasuhiro Uchimura[1], Shinji Imai[12], Naoko Egi[13], Hiroki Otani[7], Jun Udagawa[1]*

1 Division of Anatomy and Cell Biology, Department of Anatomy, Shiga University of Medical Science, Otsu, Japan, 2 Mathematical Sciences, Chalmers and University of Gothenburg, Göteborg, Sweden, 3 Multimedia Center, Shiga University of Medical Science, Otsu, Japan, 4 Department of Radiology, Shiga University of Medical Science, Otsu, Japan, 5 Biomedical MR Science Center, Shiga University of Medical Science, Shiga, Japan, 6 Department of Molecular Neuroscience Research Center, Biomedical MR Research Unit, Shiga University of Medical Science, Shiga, Japan, 7 Department of Developmental Biology, Faculty of Medicine, Shimane University, Izumo, Japan, 8 Division of Physics, Department of Fundamental Bioscience, Shiga University of Medical Science, Otsu, Japan, 9 Department of Fundamental Bioscience, Shiga University of Medical Science, Shiga, Japan, 10 Department of Surgery, Shiga University of Medical Science, Shiga, Japan, 11 Department of Anatomy II, School of Medicine, Kanazawa Medical University, Ishikawa, Japan, 12 Department of Orthopedic Surgery, Shiga University of Medical Science, Otsu, Japan, 13 Kyoto University Primate Research Institute, Inuyama, Japan

* udagawa@belle.shiga-med.ac.jp

**Data Availability Statement:** Data are available from: http://hdl.handle.net/10422/00012617.

## Abstract

Biometric ratios of the relative length of the rays in the hand have been analyzed between primate species in the light of their hand function or phylogeny. However, how relative lengths among phalanges are mechanically linked to the grasping function of primates with different locomotor behaviors remains unclear. To clarify this, we calculated cross and triple-ratios, which are related to the torque distribution, and the torque generation mode at different joint angles using the lengths of the phalanges and metacarpal bones in 52 primates belonging to 25 species. The torque exerted on the finger joint and traction force of the flexor tendons necessary for a cylindrical grip and a suspensory hand posture were calculated using the moment arm of flexor tendons measured on magnetic resonance images, and were compared among *Hylobates* spp., *Ateles* sp., and *Papio hamadryas*. Finally, the torques calculated from the model were validated by a mechanical study detecting the force exerted on the phalanx by pulling the digital flexor muscles during suspension in these three species. Canonical discriminant analysis of cross and triple-ratios classified primates almost in accordance with their current classification based on locomotor behavior. The traction force was markedly reduced with flexion of the MCP joint parallel to the torque in brachiating primates; this was notably lower in the terrestrial quadrupedal primates than in the arboreal

**Funding:** JU, KAKENHI # 21590194 (Japan Society for The Promotion of Science) https://www.jsps.go.jp/english/index.html The funders had no role in study design, data collection and analysis, decision to publish, or preparation of the manuscript.

**Competing interests:** The authors have declared that no competing interests exist.

primates at mild flexion. Our mechanical study supported these features in the torque and traction force generation efficiencies. Our results suggest that suspensory or terrestrial quadrupedal primates have hand structures that can exert more torque at a suspensory posture, or palmigrade and digitigrade locomotion, respectively. Furthermore, our study suggests availability of the cross and triple-ratios as one of the indicators to estimate the hand function from the skeletal structure.

## Introduction

Various types of biometric data have been used to analyze the relationship between hand skeletal morphology and grasping function. For instance, phalangeal curvature, which is pronounced in orangutans, gibbons, and chimpanzees, is an adaptive response to the habitual stresses associated with suspensory or climbing behavior in more arboreal primates [1, 2]. Regarding biometric ratios of relative lengths of different rays, the opposability index, namely, *total length of the thumb × 100/total length of the index finger*, is one of the indicators associated with manipulative skills. The opposability index is the highest in *H. sapiens* (mean, 60), followed by baboons (*Papio* spp.) and the mandrill (*Mandrillus sphynx*) (mean, 57–58). In contrast, the index is the lowest in orangutans (*Pongo* spp.) (mean, 40) [3]. Similarly, length of the thumb (total length of the metacarpal bone and proximal phalanx) relative to that of the third ray (total length of the metacarpal bone and proximal and middle phalanges) was larger in modern humans and archaic humans than in apes; therefore, these features appear to facilitate forceful precision grip and pinch grips [2]. On the other hand, the phalangeal index, namely, length of three phalanges in digit III × 100/total hand length, was found to be 42%–44% in *Papio spp.* and *T. gelada* and 52%–58% in primates from an arboreal milieu, suggesting that it may reflect the degree of hand adaptation for grasping and climbing in an arboreal milieu or walking and running in terrestrial ones [3]. Recently, Almécija et al. (2015) reported that the thumb-to-fourth-ray ratio in *H. sapiens*, excluding the distal phalanx in the fourth ray, has changed little since the evolution of the last common ancestor of *H. sapiens* and chimpanzees (*Pan troglodytes*) [4]. They further stated that this ratio in *H. sapiens* is similar to that of highly dexterous anthropoids, such as capuchins (*Sapajus apella* and *Cebus albifrons*), geladas (*Theropthecus gelada*), and *Papio* spp., but not *Pongo* spp. and *P. troglodytes*, which are less adept in the precision grip than *Papio* spp. [3–6]. Thus, the phalangeal curvature and thumb-to-digit ratios reflect the mode of grasping function and locomotion. Meanwhile, the intrinsic proportions of rays 2 to 5, namely, the metacarpal length relative to the total ray length and the proximal, middle, or distal phalangeal length relative to the total length, may be largely involved in the functioning of the primate hand. Actually, relatively short proximal and middle phalanges in terrestrial primates and African apes (*Pan* and *Gorilla*) imply an adaptation that attenuates bending moments on the fingers during quadrupedal locomotion on the ground [1]. Nevertheless, the following remains unclear 1) how the intrinsic proportions of rays affects the prehensile mode in terms of the torque distribution, 2) how the torque is altered by changes in the joint angle during prehension, and 3) how the torque generation property on each finger joint contributes to the prehensile function, such as a cylindrical grip or suspensory hand posture.

To address these questions, we introduced the cross- and triple-ratio [7], which is calculated from the lengths of the metacarpal bone, and the proximal, middle, and distal phalanges, as parameters associated with the intrinsic proportions of rays. For the first question, we

examined whether or not the primate classification using these ratios was consistent with the classification based on their behavior, i.e., arboreality, semi-arboreality, or terrestriality, which is involved in prehensile function. We also derived the mathematical relationship between the cross- and triple-ratios and the joint torque to examine any correlations between the intrinsic proportions of rays and the torque distribution among the carpometacarpal (CMC), the metacarpophalangeal (MCP) and proximal interphalangeal (PIP) joints. Regarding the second and third questions, the torques on the MCP, PIP, and distal interphalangeal (DIP) joints were calculated from the moment arms of the digital flexor muscle tendons, the distances between the centers of joints, and the joint angles obtained using magnetic resonance imaging (MRI) during different prehensile modes, i.e., a cylindrical grip and a suspensory hand posture. Using this model, we compared the profile of the torque generation efficiency which varies by the change in the degree of the joint angle between suspensory and terrestrial quadrupedal primates. This allowed us to clarify the mechanical features of primate hands in terms of joint torque generation. To verify our hypothesis that the hand musculoskeletal structures, including the intrinsic proportion of rays, are optimized to the prehensile function of the hand, we measured the force exerted on the finger and calculated the torque generation profile during suspensory hand postures in the brachiating and terrestrial quadrupedal primates using their fixed forelimb samples. We demonstrated the mechanical advantages of the arboreal and terrestrial hand during a suspensory hand posture, and quadrupedal locomotion with respect to the torque generation profile using a model and a mechanical experiment.

## Materials and methods

### Bone specimens and Computed Tomography (CT) images

We measured the bone lengths of primate specimens preserved at Gothenburg Museum of Natural History; Natural History Museum of Denmark; National Museum of Nature and Science, Tokyo; Primate Research Institute (PRI), Kyoto University; and Shimane University. We also used a collection of CT images of primates from the Digital Morphology Museum provided by the PRI. We borrowed preserved samples of one pileated gibbon (*Hylobates pileatus*), two gibbons (*Hylobates* spp.; unclassified), one spider monkey (*Ateles* sp.), and three hamadryas baboons (*P. hamadryas*) from the PRI collection. Hand samples of four crab-eating macaques (*Macaca fascicularis*) were carefully transferred from other ophthalmic experiments, which were approved by the Institutional Review Board of Shiga University of Medical Science Animal Care and Use Committee (2012-5-5 and 2014-9-9) and conducted in this center, after the completion of those ophthalmic experiments (See S1 Appendix). Hand CT images of these primates were acquired using a CT scanner (Aquilion ONE CT scanner; Toshiba Medical Systems Corporation, Tokyo, Japan). The species examined in this study are listed in Table 1. Locomotor behavior groups were classified according to previous reports [8–13]. *Ateles* sp., *Pongo* spp., and *Hylobates* spp. are arboreal and exhibit suspensory behavior [8], therefore we classified them as arboreal and suspensory species. *Cebus capucinus*, and *Saimiri sciureus* are arboreal, however, they use quadrupedal locomotion when navigating trees [8], so we classified them as arboreal primates that use quadrupedal locomotion. *Papio* spp. and *Theropithecus gelada* forage and travel primarily on the ground [8, 11], so we classified them as terrestrial primates. *Gorilla gorilla* use both terrestrial and arboreal habitats to feed, rest, and build their sleeping nests [8]. *Pan troglodytes* generally feed in trees for much of each day but travel on the ground between feeding sites. *Pan troglodytes* exhibit both quadrupedal and suspensory locomotion in an arboreal setting, whereas they use quadrupedal knuckle-walking locomotion on the ground [8]. However, both *Gorilla* and *Pan* sp. are considerably less suspensory than

**Table 1. Primate species measured in this study.**

| Family | SID | Species | n | B[a] | Specimen |
|---|---|---|---|---|---|
| Hominidae | 1 | *Homo sapiens* | 9 | T(BP) | bone |
| | 2 | *Gorilla gorilla* | 4 | SA | bone, CT |
| | 3 | *Pan troglodytes* | 7 | SA | bone, CT |
| | 4 | *Pongo abelii* | 1 | A(SS) | CT |
| | 5 | *Pongo pygmaeus* | 2 | A(SS) | bone, CT |
| Hylobatidae | 6 | *Hylobates agillis* | 1 | A(SS) | CT |
| | 7 | *Hylobates lar* | 2 | A(SS) | CT |
| | 8 | *Hylobates pileatus* | 1 | A(SS) | CT |
| | 9 | *Symphalangus syndactylus* | 1 | A(SS) | CT |
| | 10 | *Hylobates* sp. *(unclassified)* | 2 | A(SS) | CT |
| Cercopithecidae | 11 | *Cercopithecus diana* | 1 | A(Q) | bone |
| | 12 | *Cercopithecus neglectus* | 1 | SA | bone |
| | 13 | *Macaca cyclopis* | 1 | SA | bone |
| | 14 | *Macaca fascicularis* | 4 | A(Q) | CT |
| | 15 | *Macaca fuscata* | 4 | SA | bone, CT |
| Cercopithecidae | 16 | *Macaca mulatta* | 2 | SA | bone |
| | 17 | *Macaca nemestrina* | 1 | SA | bone |
| | 18 | *Macaca radiata* | 1 | SA | bone |
| | 19 | *Papio anubis* | 1 | T(Q) | bone |
| | 20 | *Papio hamadryas* | 4 | T(Q) | bone, CT |
| | 21 | *Theropithecus gelada* | 1 | T(Q) | bone |
| Atelidae | 22 | *Ateles paniscus* | 1 | A(SS) | bone |
| | 23 | *Ateles belzebuth* | 1 | A(SS) | bone |
| | 24 | *Ateles geoffroyi* | 1 | A(SS) | bone |
| | 25 | *Ateles* sp. *(unclassified)* | 1 | A(SS) | CT |
| Cebidae | 26 | *Cebus capucinus* | 1 | A(Q) | bone |
| | 27 | *Saimiri sciureus* | 1 | A(Q) | bone |

SID = species identification number,

[a]B = behavior, A = arboreal, BP = bipedal locomotion, Q = quadrupedal locomotion, SA = semi-arboreal, SS = suspensory species, T = terrestrial,

gibbons and orangutans [8, 12, 13], so we classified *Gorilla gorilla* and *Pan troglodytes* as semi-arboreal primates, but not suspensory species. *Cercopithecus neglectus* move primarily through the understory and on the ground: therefore, we classified them as semi-arboreal primates. *Cercopithecus diana* use primarily the canopy and understory layers [9], and *Macaca fascicularis* is primarily arboreal [8, 11], therefore, we classified them as arboreal primates in this study. Other *Macaca* species examined use both arboreal and terrestrial habitats, so we classified them as semi-arboreal species.

Information about the samples is presented in Table 1 and S1 Table.

## Bone measurements

The lengths of the metacarpal bones and the proximal, middle, and distal phalanges were measured using digital calipers (Fig 1A). The lengths of the metacarpal bone and proximal and middle phalanges were measured between the head and base of the bones in a dorsal view, whereas the length of the distal phalanx was measured between the same in a lateral view because the most concave facet of the base in the lateral view of the distal phalanx was closely

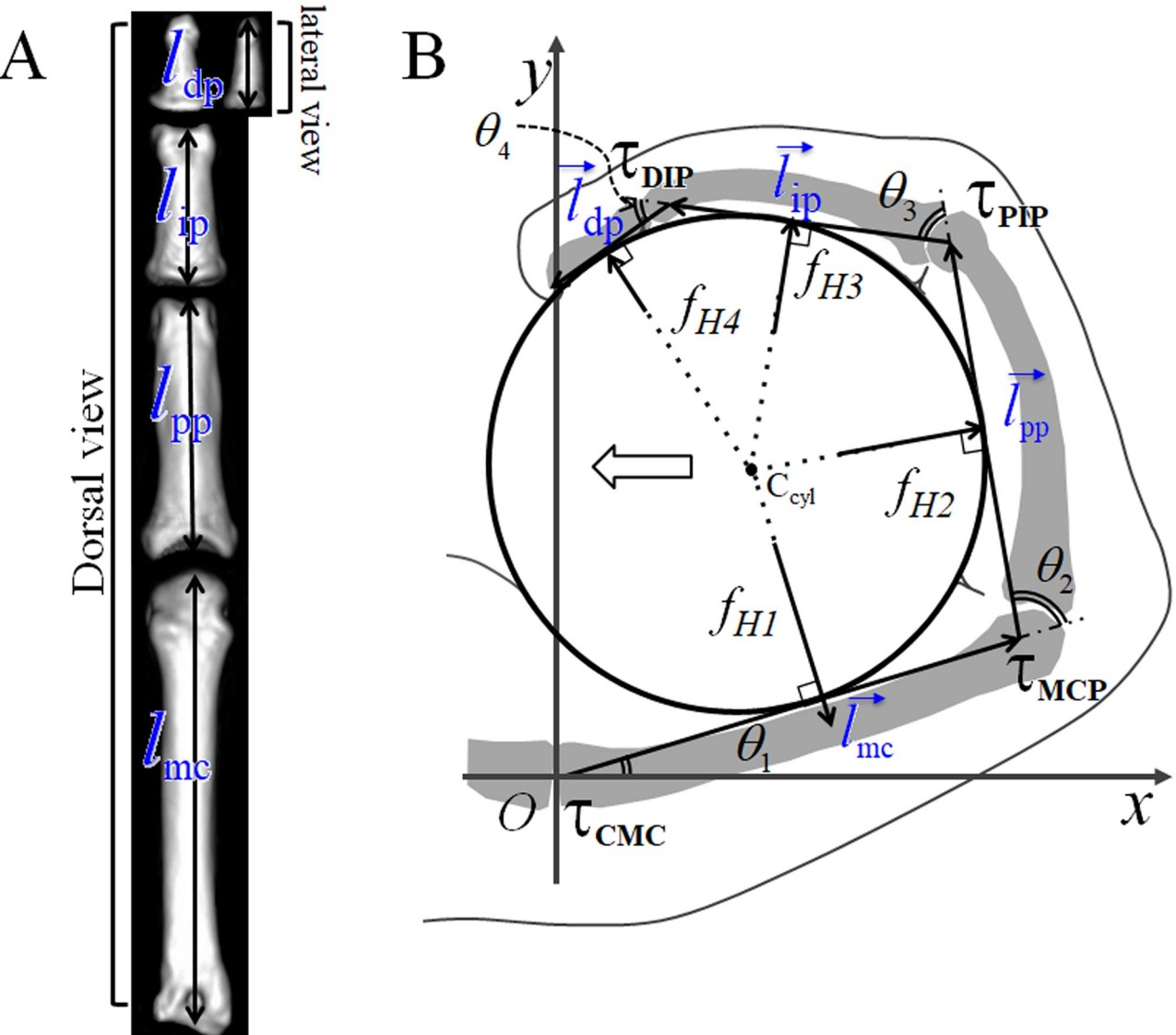

**Fig 1. Landmarks on finger bones and calculation of joint torques in the hand.** (A) CT image of the phalanges and metacarpal bone of *Papio hamadryas*: the lengths of the metacarpal bone and the proximal and middle phalanges were measured between the head and base of the bones in a dorsal view. The length of the distal phalanx was measured between them in a lateral view. (B) The holding torques of the carpometacarpal (CMC) ($\tau_{CMC}$), metacarpophalangeal (MCP) ($\tau_{MCP}$), proximal interphalangeal (PIP) ($\tau_{PIP}$), and distal interphalangeal (DIP) ($\tau_{DIP}$) were calculated using the lengths of the bones, joint angles, $\theta_1$, $\theta_2$, $\theta_3$, and $\theta_4$, and the reaction forces from the cylindrical object to the finger, $f_{H1}$, $f_{H2}$, $f_{H3}$, and $f_{H4}$. The white arrow indicates the reaction force exerted on the palm or the thumb during power grasping. $C_{cyl}$: A central axis of a cylinder.

located to the distal end of the head of the middle phalanx. Similarly, for some species, CT scan images were used to measure these lengths using the OsiriX Digital Imaging and Communication in Medicine viewer (Pixemeo SARL, Bernex, Switzerland) (Fig 1A). We calculated the cross and triple-ratios from bone lengths measured using calipers as well as reconstructed images obtained by CT scanning. The resolution of the CT images ranged from $0.125 \times 0.125$ to $0.761 \times 0.761$. It should be noted that cross and triple-ratios are dimensionless quantities that remain constant without size calibration between data acquired from the two methods

(see the next section "Cross and triple-ratios"). The length was not measured in digit III of *Pan troglodytes* #5 and digit IV of *Pan troglodytes* #5 because of the bone fracture of the distal phalanx and the metacarpal bone, respectively. The length of the distal phalanx of digit III was not measured in *Papio hamadryas* #1 because that phalanx was missing.

**Cross and triple-ratios.** A digital ray consists of a metacarpal and three phalangeal bones. A cross-ratio [14] can be used as an indicator to decompose proportions among the three variables, namely, the lengths of the three phalanges of digits II to V, into a single dimensionless value. Hence, the cross-ratio can represent the positional relationship among the CMC, MCP, PIP, and distal interphalangeal (DIP) upon finger flexion and extension. The cross-ratio of a digit was calculated as follows [7, 15]:

$$\frac{(l_{pp} + l_{ip}) \cdot (l_{ip} + l_{dp})}{l_{ip} \cdot (l_{pp} + l_{ip} + l_{dp})} \tag{1}$$

where $l_{pp}$, $l_{ip}$, and $l_{dp}$ are proximal phalangeal, middle phalangeal, and distal phalangeal lengths, respectively (Fig 1A). See S2 Table for an overview of used symbols and abbreviations.

The correlation among the three phalanges during grasp can be analyzed using Eq (1); however, a power grasp using the palm should be analyzed with the metacarpal bone. Therefore, the triple-ratio was calculated from metacarpal ($l_{mc}$), proximal phalangeal ($l_{pp}$), middle phalangeal ($l_{ip}$), and distal phalangeal ($l_{dp}$) lengths from digits II to V (Fig 1A):

$$\frac{(l_{mc} + l_{pp}) \cdot (l_{pp} + l_{ip}) \cdot (l_{ip} + l_{dp})}{l_{pp} \cdot l_{ip} \cdot (l_{mc} + l_{pp} + l_{ip} + l_{dp})}$$

The triple-ratio can be decomposed into two different cross-ratios, which is defined by the following equation:

$$\frac{(l_{mc} + l_{pp}) \cdot (l_{pp} + l_{ip}) \cdot (l_{ip} + l_{dp})}{l_{pp} \cdot l_{ip} \cdot (l_{mc} + l_{pp} + l_{ip} + l_{dp})} = \frac{(l_{pp} + l_{ip}) \cdot (l_{ip} + l_{dp})}{l_{ip} \cdot (l_{pp} + l_{ip} + l_{dp})} \cdot \frac{(l_{mc} + l_{pp}) \cdot (l_{pp} + (l_{ip} + l_{dp}))}{l_{pp} \cdot (l_{mc} + l_{pp} + (l_{ip} + l_{dp}))} \tag{2}$$
$$= Ph\ cross\ ratio \cdot MPh\ cross\ ratio$$

where the Ph cross-ratio is calculated for the distal, middle, and proximal phalanges (Ph triplet) and the MPh cross-ratio is calculated for the metacarpal bone, the proximal phalanx, and the total length of the middle and distal phalanges (MPh triplet). We compared the values of these ratios among arboreal, semi-arboreal, and terrestrial primates and examined their correlation with grasp function.

## Torque on the joints during a power grasp

To calculate the turning force (or torque) of the phalanges and metacarpus, we first assumed that the moment of inertia of the phalangeal metacarpal segment was small and that it could be neglected when measuring the power grasp. We assumed a hand position where the joint angle does not change while holding an object (e.g., when holding the branches of a tree; Fig 1B). However, holding torques in the finger joints (τ) are required under these conditions. To examine the correlation between the intrinsic proportion of rays and the holding torque, we postulated an ideal grip, wherein we disregarded the following forces: 1) friction between the surface of the object and the skin of the hand and 2) shearing of the soft skin and underlying subcutaneous tissue led by friction. Thus, we can assume that the reaction force from the object grasped per unit area is constant on the finger surface. Under these assumptions, the

resultant force ($f_{\mathrm{Hm}}$) is normal to the cylinder surface, and it is exerted at the midpoint of the bone (Fig 1B).

We calculated the holding torques of the carpometacarpal (CMC), metacarpophalangeal (MCP), proximal interphalangeal (PIP), and distal interphalangeal (DIP) joints, namely, $\tau_{\mathrm{CMC}}$, $\tau_{\mathrm{MCP}}$, $\tau_{\mathrm{PIP}}$, and $\tau_{\mathrm{DIP}}$, respectively, as follows (S2 Appendix):

$$\begin{pmatrix} \tau_{\mathrm{CMC}} \\ \tau_{\mathrm{MCP}} \\ \tau_{\mathrm{PIP}} \\ \tau_{\mathrm{DIP}} \end{pmatrix} = \begin{pmatrix} \frac{1}{2}\alpha((l_{\mathrm{mc}}^2 + l_{\mathrm{pp}}^2 + l_{\mathrm{ip}}^2 + l_{\mathrm{dp}}^2) + l_{\mathrm{mc}}\ l_{\mathrm{pp}}C_2 + l_{\mathrm{pp}}\ l_{\mathrm{ip}}C_3 + l_{mc}l_{\mathrm{ip}}\ C_{23} + l_{ip}\ l_{\mathrm{dp}}C_4 + l_{pp}l_{\mathrm{dp}}\ C_{34} + l_{\mathrm{mc}}l_{\mathrm{dp}}\ C_{234}) \\ \frac{1}{2}\alpha((l_{\mathrm{pp}}^2 + l_{\mathrm{ip}}^2 + l_{\mathrm{dp}}^2) + l_{\mathrm{pp}}\ l_{\mathrm{ip}}C_3 + l_{ip}\ l_{\mathrm{dp}}C_4 + l_{pp}l_{\mathrm{dp}}C_{34}) \\ \frac{1}{2}\alpha((l_{\mathrm{ip}}^2 + l_{\mathrm{dp}}^2) + l_{ip}\ l_{\mathrm{dp}}C_4) \\ \frac{1}{2}\alpha l_{\mathrm{dp}}^2 \end{pmatrix} \quad (3)$$

where α is a proportionality constant and $l_{\mathrm{mc}}$, $l_{\mathrm{pp}}$, $l_{\mathrm{ip}}$, and $l_{\mathrm{dp}}$ are the lengths of the metacarpal bone and the proximal, middle, and distal phalanges, respectively. Here CMC, MCP, PIP, and DIP joint angles are expressed as $\theta 1$, $\theta 2$, $\theta 3$, and $\theta 4$ (Fig 1B), and $C_{klmn}$ and $S_{klmn}$ represent cos $(\theta_k + \theta_l + \theta_m + \theta_n)$ and $\sin(\theta_k + \theta_l + \theta_m + \theta_n)$, respectively. The torque ratios of different joints in digit III during a power grasp were calculated from Eq (3).

*Pan troglodytes* and *Gorilla gorilla* travel on the ground by knuckle-walking, during which they support their forelimb on the dorsal surface of their fingers [8]. In this posture, the torque on the PIP joint is generated through contraction of the extensor digitalis muscles in the opposite direction that torque is generated during power grasping. Hence, joint torque was not considered during knuckle-walking in this study.

## Dissection of primate upper limb muscle

**Samples.** We used the fixed adult samples of three *Hylobates* spp. [species identification number (SID) 8 and 10], one *Ateles* sp. (SID 25), and three *P. hamadryas* (SID 20). Skin, subcutaneous tissue, and extensor muscles of the right or left forearm were removed, and the flexor digitorum superficialis (FDS) and flexor digitorum profundus (FDP) were exposed with their tendons without hand dissection (Fig 2A). The formalin-fixed forearm was immersed in saline containing 0.05% sodium azide so that the fingers could be easily flexed, which improved the contrast of MR images. The forearm was set in a cylinder equipped with a wooden club that was grasped by the hand by pulling FDS and FDP muscles in the forearm. The wooden club measured 2, 3, or 4 cm in diameter for *Hylobates* spp. and *Ateles* sp. and 1.5, 2, or 3 cm in diameter for *P. hamadryas* was used, depending on the size of the hand (Fig 2A).

## Magnetic resonance imaging

Sagittal images of the primate hands were acquired using the 4.7T MRI system (BioSpec 47/40 USR; Bruker, Billerica, Massachusetts) while grasping the wooden club by pulling FDS and FDP muscles. The hand was scanned in the extended position and in the flexed position while grasping wooden clubs measuring 2 and 3 cm in diameter. For *Hylobates* spp. and *Ateles* sp., the hand was also scanned in the flexed position while tightly or mildly grasping a wooden club measuring 4 cm in diameter. We obtained scans of five different flex positions, with five different DIP, PIP, and MCP angles, to improve the accuracy of the regression curve of the torque and the moment arm. Similarly, the hand of *P. hamadryas* was scanned in the extended position and the flexed position while grasping wooden clubs measuring 2 and 3 cm in

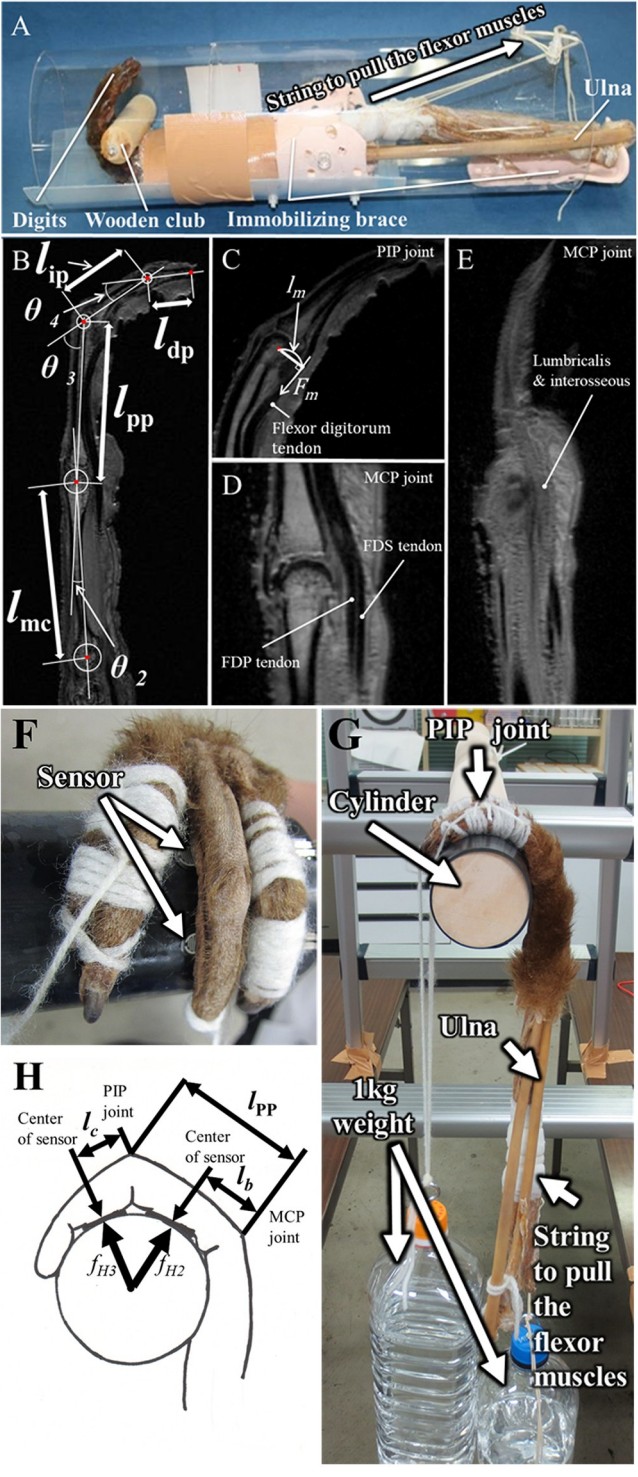

**Fig 2. Measurements of the distance between joint centers, joint angles, and moment arms during flexion of the digits, and the force exerted on the phalanges during a suspensory hand posture.** (A) The forearm was settled in the cylinder equipped with a wooden club that was grasped with hand by pulling the flexor digitorum superficialis (FDS) and flexor digitorum profundus (FDP) in the forearm. The wooden club was grasped with the primate hand, and sagittal images of the primate hands were acquired using a 4.7T MRI system. (B) The centroid of the circle superimposed on the concave proximal facet of the phalanx was defined as the joint center of rotation. The distance between the rotation centers was defined as the lengths of the metacarpal bone ($l_{mc}$), and the proximal ($l_{pp}$) and middle ($l_{ip}$) phalanges. The length of the distal phalanx ($l_{dp}$) was defined as the distance from the center of the distal

interphalangeal (DIP) joint to the tip of the distal phalanx. (C) At the proximal interphalangeal (PIP) joint, a moment arm of the flexor digitorum muscle tendons ($l_m$) was measured without discrimination of FDP and FDS tendons because both tendons pass through the palmer side of the PIP joint at almost the same distance from the axis of this joint and cannot be discriminated in MR images. The traction force of the tendon by muscle contraction is indicated as $F_m$. (D) At the metacarpophalangeal (MCP) joint, moment arms of FDP and FDS tendons were separately measured. However, (E) lumbricalis and interosseous muscles were not discriminated, therefore, a moment arm of a bundle of intrinsic muscles, which includes the lumbricalis, was measured at the radial side of the MCP joint. Pressure sensors were placed between the column and palmar surface of the proximal or middle phalanx (F). A plastic bottle filled with water (1 kg weight) was strung from the distal phalanx of digit III together with the digit II, IV, and V to fix the finger on the column wrapped with a rubber sheet (G). On the other side, another 1kg weight was strung from the digital flexor muscles to pull their tendons (G). The torques of MCP, and PIP joints were calculated using the force measured by the sensors, $f_{H2}$, and $f_{H3}$, the distance from the MCP or PIP joint to the center of the sensor, $l_b$, and $l_c$, the length of the proximal phalanx, $l_{pp}$, and PIP joint angle (H).

diameter, in addition to the flexed position while tightly or mildly grasping a wooden club measuring 1.5 cm in diameter to accommodate the smaller hand size of *P*. hamadryas. Rapid acquisition with a relaxation enhancement sequence was applied to all samples. Parameters of MRI sequences are described in S3 Table. MRI images were obtained for five different flex postures of each hand. Three-dimensional images were reconstructed from 2D images, where the bones and tendons of flexor digitorum muscles were colored using ImageJ software (imagej. nih.gov/ij/) [16] (S1 Fig).

## Calculation of torque generation efficiency

The ratios of the torque relative to the moment arm on each joint, or the traction forces by flexor muscles needed to generate a holding torque, were compared among *Hylobates* spp., *Ateles* sp., and *P*. hamadryas to examine the difference in torque generation efficiency between brachiating primates and terrestrial primates. We defined the centroid of the circle superimposed on the concave proximal facet of the phalanx as the rotation center of the joint (Fig 2B). The distance between the centers of MCP and PIP joints or those of PIP and DIP joints was defined as the length of the proximal phalanx ($l_{pp}$) or the length of the middle phalanx ($l_{ip}$). The length of the distal phalanx ($l_{dp}$) was defined as the distance from the center of DIP joints to the tip of the distal phalanx. To examine the efficiency of torque generation with the finger size effect removed, normalized load torques on MCP, PIP, and DIP joints ($\tau_{MCP\_N}$, $\tau_{PIP\_N}$, and $\tau_{DIP\_N}$, respectively) were calculated using the normalized lengths of the phalanges relative to the total length ($L = l_{pp} + l_{ip} + l_{dp}$) of the phalanges, $l_{pp}/L$, $l_{ip}/L$, $l_{dp}/L$ respectively, with Eq (3).

$$\begin{pmatrix} \tau_{MCP_N} \\ \tau_{PIP_N} \\ \tau_{DIP_N} \end{pmatrix} = \begin{pmatrix} \frac{1}{2}\alpha\left(\left(\frac{l_{pp}^2}{L^2}+\frac{l_{ip}^2}{L^2}+\frac{l_{dp}^2}{L^2}\right)+\frac{l_{pp}\,l_{ip}}{L^2}C_3+\frac{l_{ip}\,l_{dp}}{L^2}C_4+\frac{l_{pp}l_{dp}}{L^2}C_{34}\right) \\ \frac{1}{2}\alpha\left(\left(\frac{l_{ip}^2}{L^2}+\frac{l_{dp}^2}{L^2}\right)+\frac{l_{ip}\,l_{dp}}{L^2}C_4\right) \\ \frac{1}{2}\alpha\frac{l_{dp}^2}{L^2} \end{pmatrix} \quad (4)$$

In the suspensory hand postures of *Hylobates* and *Ateles* sp., the PIP joint was positioned on the top of the support [13]; therefore, the holding torques of MCP and PIP joints during

suspensory hand postures were calculated as follows (S3 Appendix):

$$\begin{pmatrix} \tau_{\text{MCP}} \\ \tau_{\text{PIP}} \end{pmatrix} = \begin{pmatrix} \frac{1}{2} f_u r^2 (\sin 2\beta + \sin (2\gamma + 2\delta))^2 \\ \frac{1}{2} f_u r^2 \sin^2 (2\gamma + 2\delta) \end{pmatrix} \qquad (5)$$

In this posture, it is suggested that the force is mainly loaded to the proximal and middle phalanges; hence, we calculated the alternative torque $\tau_n^{\#}$, where the contribution of the distal phalanx was eliminated.

$$\begin{pmatrix} \tau_{\text{MCP}}^{\#} \\ \tau_{\text{PIP}}^{\#} \end{pmatrix} = \begin{pmatrix} \frac{1}{2} f_u r^2 (\sin 2\beta + \sin 2\gamma)^2 \\ \frac{1}{2} f_u r^2 \sin^2 2\gamma \end{pmatrix} \qquad (6)$$

Here, $f_u$ is force per unit length, $r$ is the diameter of the circle on which the rotation centers of MCP, PIP, and DIP joints lie, and $2\beta$ and $2\gamma$ are central angles of the intercepted arcs, which are formed between the rotation centers of MCP and PIP joints and between those of PIP and DIP joints, respectively.

The moment arm of the FDP tendon was measured at the DIP joint, but at the PIP joint, the moment arm of the flexor digitorum muscle tendons was measured without discrimination of FDP and FDS tendons. This is because both tendons pass through the palmar side of the PIP joint at almost the same distance from the axis of this joint, and they cannot be discriminated in MRI (Fig 2C). At the MCP joint, the moment arms of FDP and FDS tendons were discriminated and measured at the midline of each finger (Fig 2D); however, the lumbricalis and interosseous muscles were not discriminated (Fig 2E). Hence, the moment arm of a bundle of intrinsic muscles, which included the lumbricalis, was measured at the radial side of the MCP joint (Fig 2E). Joint torque was generated by the traction of the tendon through the pulling muscle. Normalized torque ($\tau_N$), which is calculated using Eqs 4 or 6, is described by the following equation:

$$\tau_N = F_m \times \frac{l_m}{|L|} \qquad (7)$$

Here, $F_m$ and $l_m$ are the traction forces of the tendon and moment arm, respectively (Fig 2). Then, $F_m$ was calculated as follows:

$$F_m = \tau_N \cdot \frac{|L|}{|l_m|} \qquad (8)$$

Therefore, if the torque is constant and $F_m$ is small, the efficiency of torque generation on the joint is high. Correlations between the joint angle and $F_m$ or $l_m$ were examined at DIP, PIP, and MCP joints. The joint angle was measured, and $\tau_N$ and $F_m$ were calculated at five flex postures in each individual. This resulted in 15 data points for each joint, which were used for the correlation analysis between the joint angle and $\tau_N$ or $F_m$ by combining data from three individuals (*P. hamadryas* or *Hylobates*).

## A mechanical study: Measurement of the force exerted on the phalanx during suspensory hand posture

Force exerted on the proximal and middle phalanges was measured using resistor-type pressure sensors measuring 5.08 mm in diameter (FSR$^{®}$ 400; Interlink Electronics, Camarillo, CA) in the forelimb samples of *P. hamadryas* and *Hylobates spp*. Sensors were placed between the palmar surface of the proximal and middle phalanges and the column, so that resistance could be measured adequately (Fig 2F). The electrical resistance was measured with a digital tester (TDX-200; Ohm Electric, Saitama, Japan) in a suspensory hand posture where a 1 kg weight was strung from the distal phalanx of digit III together with digits II, IV, and V to fix the fingers over the column wrapped with a rubber sheet, or to stop the fingers from slipping down. On the opposite side, another 1 kg weight was strung from the digital flexor muscles to pull their tendons downwards with a stable force (9.8 N) (Fig 2G). We used the columns measuring 2, 3, 4, 5, or 8 cm in diameter for *Hylobates* spp.; and 1.5, 2, 3, 4, or 5 cm in diameter for *P. hamadryas* depending on the size of the hand. The calibration curve for resistance-force conversion was acquired using weights over the range of 30g to 1,150g. The determinant coefficient of the calibration curve was 0.99 using a power function as a fitting curve. Torques generated on the PIP and MCP were compared among different joint angles using the equations in S2 Appendix because the force normal to the longitudinal axis of the phalanx was measured by the pressure sensor (Fig 2H):

$$\begin{pmatrix} \tau^{\#}_{\mathrm{MCP}} \\ \tau^{\#}_{\mathrm{PIP}} \end{pmatrix} = \begin{pmatrix} l_b f_{H2} + (l_{\mathrm{pp}} C_3 + l_c) f_{H3} \\ l_c f_{H3} \end{pmatrix} \quad (9)$$

where $l_{\mathrm{pp}}$ is length of the proximal phalanx, $l_b$ is the distance between the MCP joint and the center of the pressure sensor on the proximal phalanx, $l_c$ is the distance between the PIP joint and the center of the pressure sensor on the middle phalanx, and $F_{H2}$ or $f_{H3}$ is a reaction force exerted on the proximal or middle phalanx (Fig 2H).

Data were acquired from three *Hylobates*, and three *P*. hamadryas, whereas sufficient data could not be acquired from one *P*. hamadryas when the columns measuring 5 cm was held because of incompatibility between the finger and the columns or the sensors, due to their thinner and shorter finger. Torque data were corrected using the total weight including the weights (2 kg) and the forearm weight as the following:

$$\mathrm{Corrected\ torque} = \mathrm{measured\ torque} \times \frac{weights(2\ kg)}{weights(2\ kg) + forearm\ weight}$$

The torque-PIP joint angle curve profile represents the torque output properties with respect to different degrees of finger flexion. The curve profile, but not the absolute value of the torque, may be independent of individual body mass during a suspensory hand posture because the magnitude of $F_{H2}$ or $f_{H3}$ in Eq 9 is proportional to the body mass. Hence, the torque-PIP joint angle curve profiles of *Hylobates* and *P. hamadryas* were compared to examine the difference in torque output properties on the MCP and PIP joints between a suspensory species and a terrestrial quadrupedal primate.

## Torque on the joints during quadrupedal locomotion on tree branches

The hand posture used by primates during quadrupedal locomotion on branches is palmigrady, in which the primary loading area is the palm [17, 18] Torque on the joints during the adoption of a palmigrady posture on vertical substrates can be calculated using the equation

for power grasp as described above; therefore, that on horizontal substrates was calculated (S5 Appendix and S8 Fig). *Macaca fascicularis*, *Cercopithecus diana*, *Cebus capucinus*, and *Saimiri sciureus* were chosen as the quadrupedal primates in the arboreal group, and *Papio anubis*, *Papio hamadryas*, and *Theropithecus gelada* were chosen as the terrestrial quadrupedal primates in our samples. Calculated torques on the MCP and PIP joints were compared between the arboreal and terrestrial quadrupedal primates.

## Statistics

We used R3.2.4 and SPSS Statistics version 22 to analyze the data. Primate species were classified based on hand cross or triple-ratios using canonical discriminant analysis (CDA) and finite mixture analysis (FMA). Multivariate classification of primate species was also performed by CDA with four variables of triple-ratios and eight variables of Ph and MPh cross-ratios in digits II to V; the most important discriminant variable among these ratios was determined. FMA was also applied to the classification of primates in this study because it is highly accurate when used on datasets with small sample sizes and missing data. FMA also accurately assigns a status of "unknown" to taxa [19]; therefore, mclust, an R package for FMA was used for classification in this study. After classifying primates into clusters using CDA and FMA, two-way repeated measures analysis of variance was used to analyze the interaction between the primate clusters and the cross or triple-ratio from digits II to V using SPSS. This allowed us to examine any differences in the patterns of the cross and triple-ratios of these digits between the clusters and between humans and terrestrial quadrupedal primates. Torque ratios of the PIP, MCP, and CMC joints were compared among the clusters using two-way analysis of variance and any interspecific differences were analyzed using Scheffe's *post hoc* test [20] in SPSS. Differences in $\tau_N$ and $F_m$ were compared between *P. hamadryas* and *Hylobates* by generalized estimating equations (GEE) using geepack in R. Differences were considered significant at $P < 0.05$.

## Results

### Primate classification based on hand cross and triple-ratios

The mean triple or cross-ratio in each species was used for CDA. According to CDA using the triple and cross-ratios of the hands, the primates were almost adequately clustered into the arboreal, semi-arboreal, and terrestrial groups (p < 0.001) (Fig 3A and 3B, and Table 2). The gorilla (*Gorilla gorilla*) was clustered into the semi-arboreal group, and it is a relatively good tree climber despite rarely using suspensory feeding postures [8, 21]; hence, this classification may reflect the gorilla's hand f*unctio*n. In the reclassification of primate species, 96% of the primates, with the exception of *M. fascicularis*, were correctly classified by CDA using both triple and cross-ratios (Fig 3A and 3B, and Table 3). However, *M. fascicularis* is primarily arboreal but is also adept on terrestrial substrates; therefore, CDA using these ratios may classify the primate based on its locomotor behavior.

In FMA, raw data of the triple-ratio in every sample were used; however, the results were similar to those of CDA, although the white-headed capuchin (*Cebus capucinus*) and the Diana monkey (*Cercopithecus diana*) were classified within the semi-arboreal cluster (Fig 3C). On the other hand, Hominoidea species (*Pongo* spp., *Hylobates* spp., *P. troglodytes*, *G. gorilla*, and *H. sapiens*) were classified into three groups, arboreal, semi-arboreal, and terrestrial clusters, in both CDA and FMA (Fig 3). *H. sapiens* was grouped with Old World monkeys in both CDA and FMA (*Papio* spp. and *T. gelada*; Fig 3); however, *C. capucinus*, which is also a New

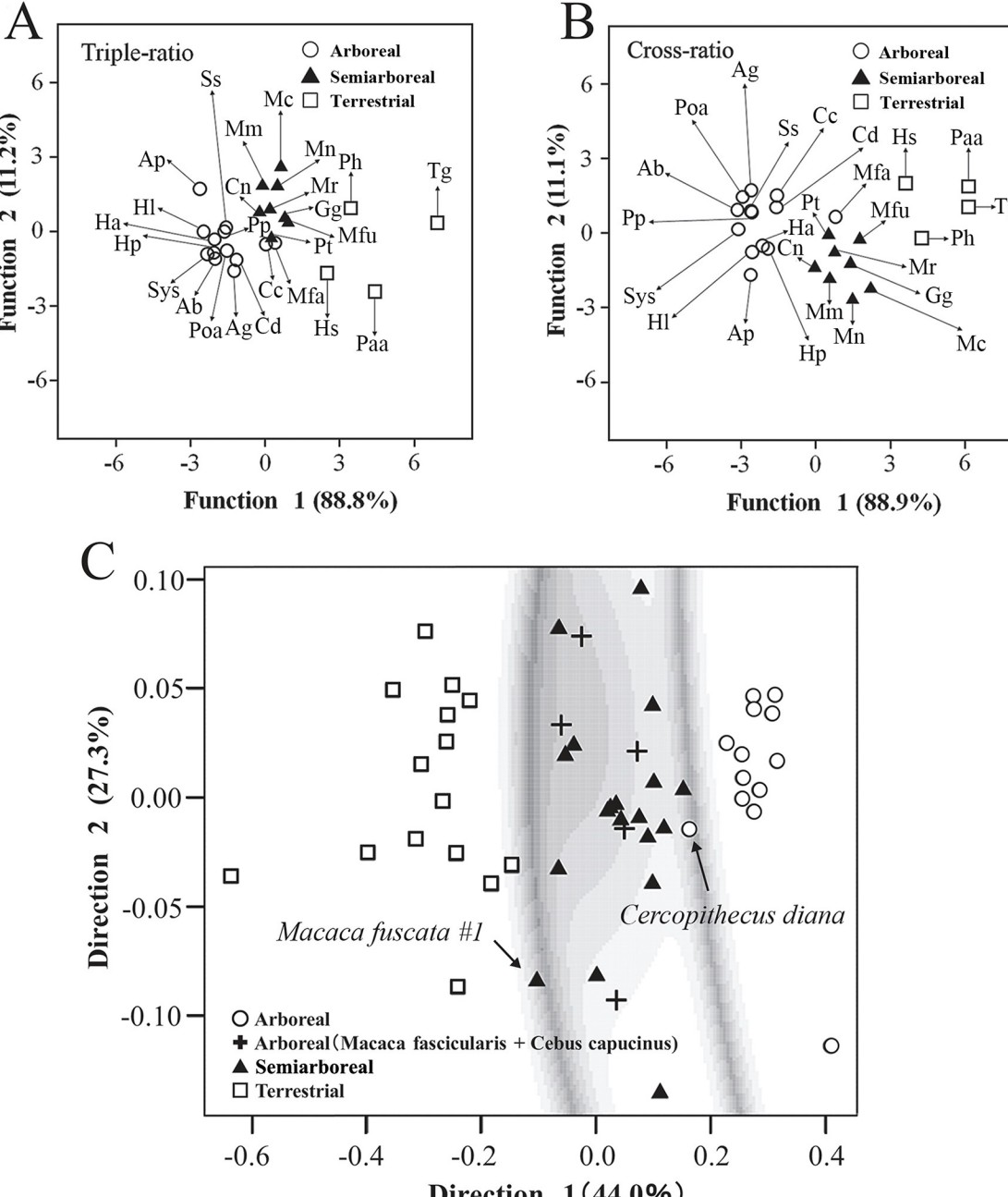

**Fig 3. Individual samples scores of the three examined groups obtained by canonical discriminant analysis on the two canonical functions using triple- and cross-ratios, and a finite mixture analysis.** A multivariate classification of primate species was performed by canonical discriminant analysis with four variables of a triple-ratio (A) or eight variables of Ph and MPh cross-ratios (B) in *digits* II to V. Additionally, a finite mixture analysis using four variables of a triple-ratio (C) was applied to primate classification. Uncertainty classification boundaries were shown as darker greyscale region (C). Ss. *Saimiri sciureus*, Pp. *Pongo pygmaeus*, Mfa. *Macaca fascicularis*, Cc. *Cebus capucinus*, Cd. *Cercopithecus diana*, Ag. *Ateles geoffroyi*, Poa. *Pongo abelii*, Ab. *Ateles belzebuth*, Sys. *Symphalangus syndactylus*, Hp. *Hylobates pileatus*, Ha. *Hylobates agillis*, Hl. *Hylobates lar*, Ap. *Ateles paniscus*, Mm. *Macaca mulatta*, Mc. *Macaca cyclopis*, Mn. *Macaca nemestrina*, Mr. *Macaca radiate*, Gg. *Gorilla gorilla*, Mfu. *Macaca fuscata*, Pt. *Pan troglodytes*, Cn. *Cercopithecus neglectus*, Ph. *Papio hamadryas*, Tg. *Theropithecus gelada*, Paa. *Papio anubis*, and Hs. *Homo sapiens*.

**Table 2. Eigen values and test of significance of the discriminant functions in CDA.**

| | Triple-ratio | | Cross-ratio | |
|---|---|---|---|---|
| Function | 1 | 2 | 1 | 2 |
| Eigen valued | 4.884 | 0.617 | 7.902 | 0.990 |
| Percent of variance | 88.8 | 11.2 | 88.9 | 11.1 |
| Canonical correlation | 0.911 | 0.618 | 0.942 | 0.705 |
| | Triple-ratio | | Cross-ratio | |
| Test of Functions | 1 through 2 | 2 | 1 through 2 | 2 |
| Wilks' Lambda | 0.105 | 0.618 | 0.056 | 0.502 |
| Chi-square | 46.186 | 9.855 | 53.180 | 12.734 |
| df | 8 | 3 | 16 | 7 |
| $p$ value | < 0.001 | 0.020 | < 0.001 | 0.079 |

World monkey, was clustered with Old World monkeys in FMA (Macaque and Cercopithecidae; Fig 3C). The results of CDA and FMA based on the cross or triple-ratio were found to be consistent with locomotor behavior or hand function.

To clarify which digit is important for discriminating the primates, we examined the canonical function in CDA. Most of the variation was explained by the first canonical function, since the contributions of the first axis were 88.8% and 88.9% using triple and cross-ratios, respectively (Table 2). Therefore, three groups were distinguished primarily by the variable in the first function. In the first CDA function using the triple-ratio, digit III (0.714) was the most important discriminant variable, followed by digits II (0.250), V (0.111), and IV (0.028) (Table 4). Similarly, digit III (1.276) was the most important discriminant variable in the first function, followed by MPh (0.666) and Ph (−0.504) of digit II in CDA using the cross-ratio (Table 4).

## Differences between cross and triple-ratios of digits II to V among the clusters classified by CDA and a finite mixture model

The hand triple-ratio and Ph and MPh cross-ratios were compared among arboreal (cluster 1; n = 10; SID 4–9, 22–24, and 27), semi-arboreal (cluster 2; n = 9: SID 2, 3, and 12–18), and terrestrial clusters (cluster 3; n = 4; SID 1 and 19–21) (Table 1), which consisted of the species common to both classifications using CDA and a finite mixture model. They differed significantly among clusters 1, 2, and 3, and the values were smallest in the arboreal cluster and largest in the terrestrial cluster (Fig 4A, 4B and 4C). The hand triple-ratio and MPh cross-ratio significantly differed between *H. sapiens* and terrestrial quadrupedal primates (Fig 4D and 4E), but there was no significant difference in the Ph cross-ratio between them (Fig 4F).

**Table 3. Classification results by CDA using the triple- or cross-ratio.**

| | Predicted Group Membership | | | | Correctly classified (%) |
|---|---|---|---|---|---|
| | Arboreal | Semi -arboreal | Terrestreal | Total | |
| Arboreal | 12 | 1 | | 13 | 92.3 |
| Semi-arboreal | | 8 | | 8 | 100.0 |
| Terrestrial | | | 4 | 4 | 100.0 |
| Total | | | | 25 | 96.0 |

**Table 4. Standardized coefficients in canonical discriminant functions.**

| Variable | | Function 1 | Function 2 |
|---|---|---|---|
| Triple-ratio | digit II | 0.250 | −0.351 |
| | digit III | 0.714 | −0.032 |
| | digit IV | 0.028 | 1.562 |
| | digit V | 0.111 | −1.177 |
| Ph CR[a] | digit II | −0.504 | 0.797 |
| | digit III | 1.276 | −0.161 |
| | digit IV | 0.245 | −0.525 |
| | digit V | −0.128 | 0.844 |
| MPh CR[b] | digit II | 0.666 | 0.651 |
| | digit III | 0.261 | −0.111 |
| | digit IV | −0.244 | −0.740 |
| | digit V | 0.403 | −0.122 |

[a]Cross ratio calculated from the proximal, middle, and distal phalangeal lengths

[b]Cross ratio calculated from the metacarpal, and proximal, middle, and phalangeal lengths

## Correlation between cross and triple-ratios and joint torque distribution

**Cylindrical grip and suspensory hand posture.** The Ph and MPh ratio and triple-ratios can be described by the square root of the torque distribution in a cylindrical grip (S4 Appendix). Additionally, the cross and triple-ratios of digit III were the most important variables in CDA (Table 4); hence, the torque distributions of the finger joints were calculated in digit III using Eq (4) in arboreal (n = 10; SID 4–9, 22–24, and 27), semi-arboreal (n = 9: SID 2, 3, and 12–18), and terrestrial primates (n = 4; SID 1 and 19–21) (Fig 5). $\sqrt{\frac{\tau_{PIP}}{\tau_{MCP}}}$ or $\sqrt{\frac{\tau_{MCP}^{\#}}{\tau_{PIP}^{\#}}}$ during a cylindrical grip was significantly larger in terrestrial primates at extension and mild flexion ($r \geq 0.32$) or at all positions examined in this study, respectively, than in arboreal primates (Fig 5A and 5B). On the other hand, the $\sqrt{\frac{\tau_{MCP}}{\tau_{CMC}}}$ value was smallest in terrestrial quadrupedal monkeys, although the $\sqrt{\frac{\tau_{PIP}}{\tau_{CMC}}}$ value did not differ between arboreal and terrestrial quadrupedal primates when $r$ ranged between 0.28 (at flexion) and 0.50 (at extension) (Fig 5C and 5D). A larger $\sqrt{\frac{\tau_{PIP}}{\tau_{MCP}}}$ and smaller $\sqrt{\frac{\tau_{MCP}}{\tau_{CMC}}}$ at extension and mild flexion reflect a smaller torque load on the MCP joint in terrestrial primates during quadrupedal locomotion.

Interestingly, $\sqrt{\frac{\tau_{PIP}}{\tau_{MCP}}}$ during suspensory hand posture was significantly larger upon strong flexion of the finger ($r = 0.28$ and 0.30) but smaller upon mild flexion ($r = 0.40$ and 0.50) in the brachiating primates (n = 9; *Hylobates* spp., *Ateles* spp., and *Pongo* spp.) than in terrestrial quadrupedal primates (n = 3) (Fig 5E). $\sqrt{\frac{\tau_{MCP}^{\#}}{\tau_{PIP}^{\#}}}$ was considerably smaller in brachiating primates during suspensory hand posture as well as in arboreal primates during a cylindrical grip than in terrestrial quadrupedal primates (Fig 5F). Smaller $\sqrt{\frac{\tau_{MCP}^{\#}}{\tau_{PIP}^{\#}}}$ and larger $\sqrt{\frac{\tau_{PIP}}{\tau_{MCP}}}$ values in brachiating primates during strong flexion of the finger may indicate that a smaller torque distribution on the MCP joint may be advantageous in suspensory hand postures. Taken together, our results suggest that load is less distributed on the MCP joint in brachiating

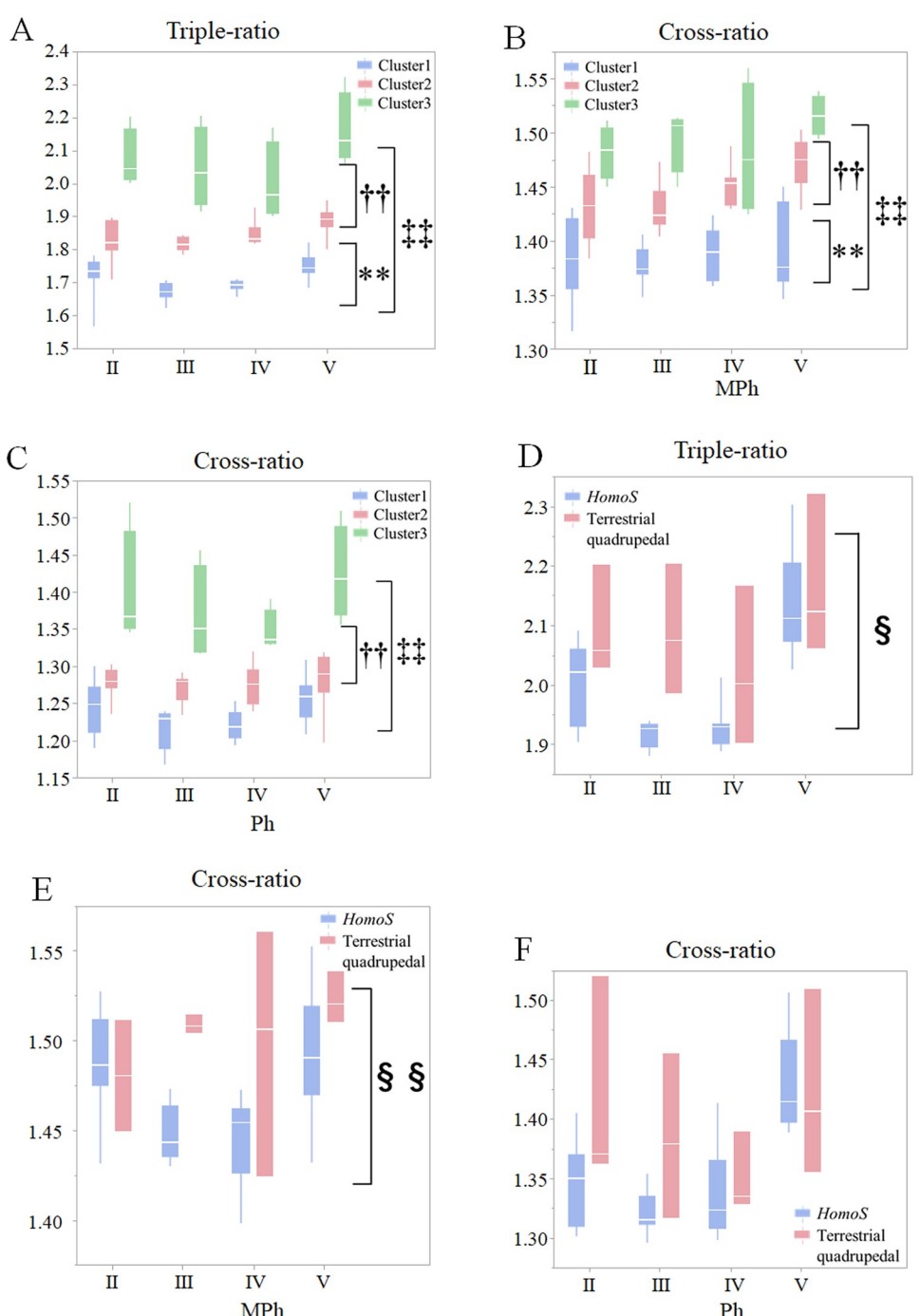

**Fig 4. Comparisons of triple- and cross-ratios in the hand skeletons of primates.** Box plots of the triple- and cross-ratios of the hand skeletal structures. The triple-ratio (A), MPh cross-ratio (B), and Ph cross-ratio (C) was compared among primates grouped into cluster 1, 2, or 3, and the triple-ratio (D), MPh cross-ratio (E), and Ph cross-ratio (F) between *Homo sapiens* and terrestrial quadrupedal monkeys in digits II to V using Repeated Measures ANOVA. White lines within box plots are median values. $^{*}P < 0.05$, $^{**}P < 0.01$, cluster 1 versus cluster 2;$^{\ddagger}P < 0.05$, $^{\ddagger\ddagger}P < 0.01$, cluster 1 versus cluster 3; $^{\dagger}P < 0.05$, $^{\dagger\dagger}P < 0.01$, cluster 2 versus cluster 3; $^{\S}P < 0.05$, $^{\S\S}P < 0.01$, humans versus terrestrial monkeys using Scheffe's *post hoc* test.

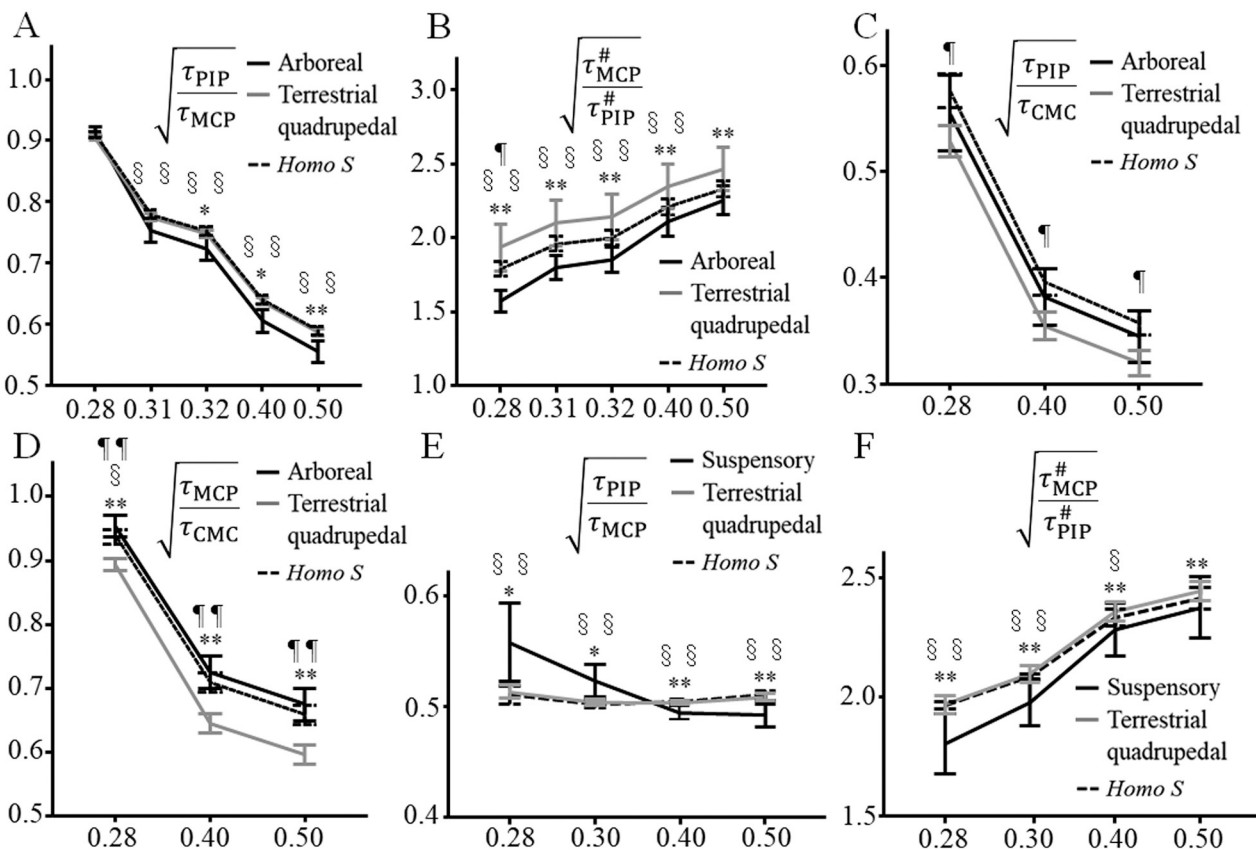

**Fig 5. Torque ratios among proximal interphalangeal (PIP) and metacarpophalangeal (MCP), and carpometacarpal (CMC) joints in digit III during a cylindrical grip or suspensory hand posture.** $\sqrt{\frac{\tau_{PIP}}{\tau_{MCP}}}$ (A), $\sqrt{\frac{\tau_{MCP}^{\#}}{\tau_{PIP}^{\#}}}$ (B), $\sqrt{\frac{\tau_{PIP}}{\tau_{CMC}}}$ (C), and $\sqrt{\frac{\tau_{MCP}}{\tau_{CMC}}}$ (D) during cylindrical grip are compared among the arboreal group, terrestrial quadrupedal group, and *Homo sapiens*. $\sqrt{\frac{\tau_{PIP}}{\tau_{MCP}}}$ (E), and $\sqrt{\frac{\tau_{MCP}^{\#}}{\tau_{PIP}^{\#}}}$ (F) during suspensory hand posture was also compared among brachiating, and terrestrial quadrupedal primates, and *H. sapiens*. Ph cross-ratio, the cross-ratio calculated from the proximal, middle, and distal phalangeal lengths; $r_I$, diameter of the cylinder relative to the sum of the lengths of the proximal, middle, and distal phalanges; $\tau_{CMC}$, $\tau_{MCP}$ and the $\tau_{PIP}$ torques loaded on the CMC, MCP and PIP joints, respectively; $\tau_{MCP}^{\#}$ and $\tau_{PIP}^{\#}$ the torques loaded on the MCP and PIP joints, respectively, when the force loaded on the distal phalanx is not considered. $^{*}P < 0.05$, $^{**}P < 0.01$, arboreal or brachiating primates versus terrestrial quadrupedal; $^{\S}P < 0.05$, $^{\S\S}P < 0.01$, arboreal or brachiating primates versus *H. sapiens*, $^{\P}P < 0.05$, $^{\P\P}P < 0.01$, terrestrial quadrupedal versus *H. sapiens*, using Scheffe's *post hoc* test.

primates during suspensory hand posture or in terrestrial quadrupedal primates during terrestrial locomotion.

## Torque and torque generation efficiency during a cylindrical grip and suspensory hand posture using our model in *Hylobates* spp., *Ateles* sp., and hamadryas

The orientation of flexor muscle tendons is closely related to the magnitude of the torque on the joint as well as the bone structures. The positional relationship between the tendons and bones in the hand changed with finger flexion (identified by MRI; S1 Fig). There was a moderate correlation between the joint angle and normalized moment arms ($l_m/L$) of flexor tendons at the PIP joint in digits III in *P. hamadryas* but not in *Hylobates* spp., and *Ateles* sp., whereas the correlation was weak between the joint angle and $l_m/L$ of flexor tendons at the DIP joint (Fig 6, S5 Fig, and S4 Table). Moreover, $l_m/L$ of FDP, FDS, and intrinsic muscle tendons at the

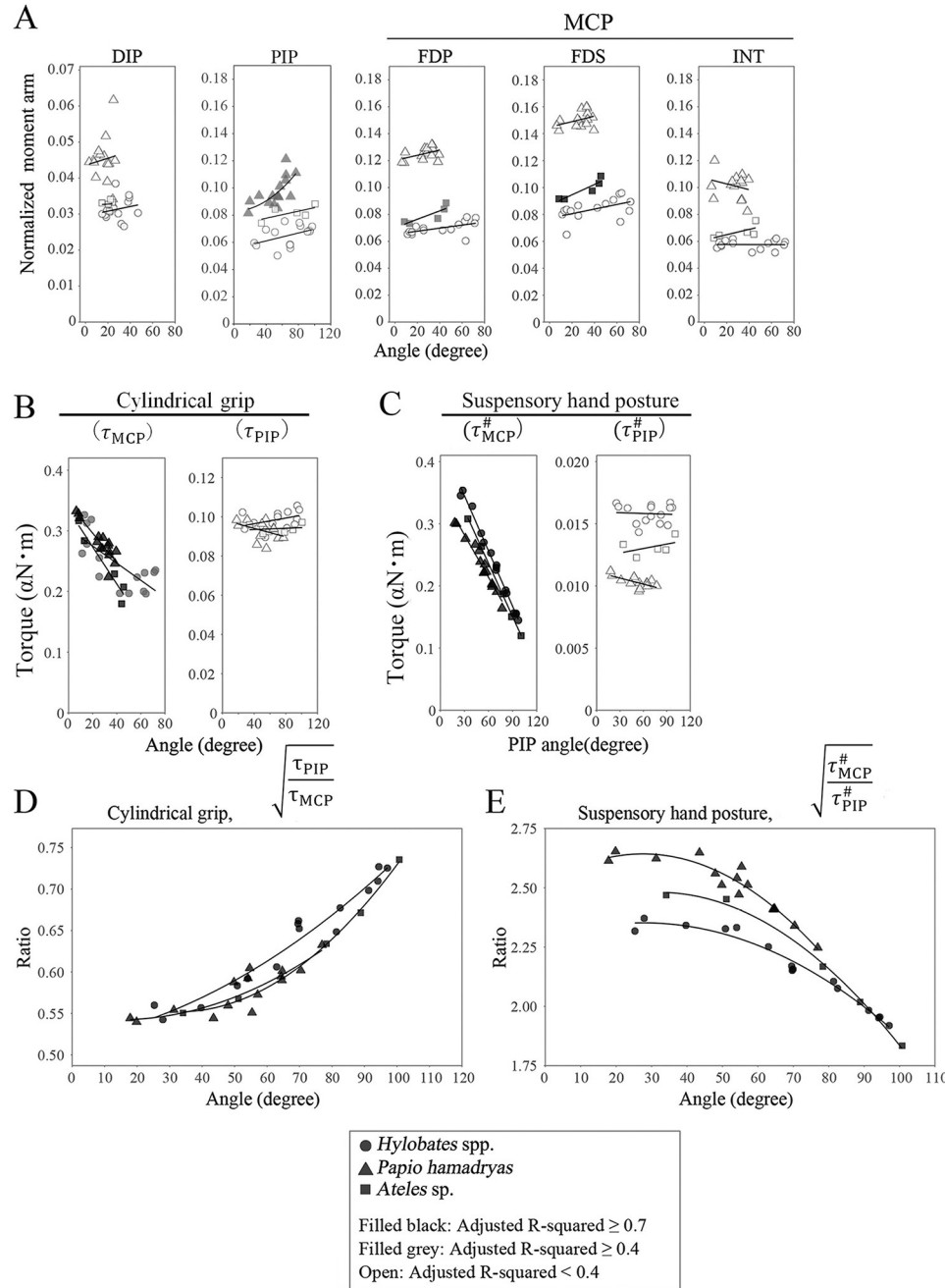

**Fig 6. Correlation between the finger joint angle and the normalized moment arm, torque, or torque ratio calculated using MR images of digit III during a cylindrical grip and suspensory hand posture.** The differences in $\frac{l_m}{L}$ (A), $\tau_{MCP}$ and $\tau_{PIP}$ (B)$\tau_{MCP}^{\#}$, and $\tau_{PIP}^{\#}$ (C), and $\sqrt{\frac{\tau_{PIP}}{\tau_{MCP}}}$ (D), $\sqrt{\frac{\tau_{MCP}^{\#}}{\tau_{PIP}^{\#}}}$ (E), were compared among *Hylobates* spp., *Papio hamadryas*, and *Ateles* sp. using generalized estimating equations (GEE). Circles, triangles, and squares show the data of *Hylobates* spp., *Papio hamadryas*, and *Ateles* sp., respectively. Symbols filled with black, $R^2$ of the regression $\geq 0.7$; symbol filled with gray, $R^2$ of the regression $\geq 0.4$; open symbol, $R^2$ of the regression $< 0.4$. Regression equations are shown in S4, S5 and S6 Tables.

MCP joint did not correlate with the joint angle in digit III of *P. hamadryas* and *Hylobates* spp., whereas $l_m/L$ values of FDP and FDS were positively correlated with the joint angle in *Ateles* sp. (Fig 6, S4 Table). By contrast, there were positive correlations between $l_m/L$ of FDP or FDS with the MCP joint angle in digits IV and V in *Hylobates* spp. and digits IV, V, and II in *Ateles* spp. (S5 Fig and S4 Table). Our results regarding the normalized moment arms suggest that the efficiency of conversion of the muscle constriction power into torque generation at the MCP joint increases with flexion of digits IV and V but not digits II and III in brachiating primates and that at the PIP joint increases with flexion of digits III in terrestrial primates, whereas it does not increase at the DIP joint in brachiating and terrestrial primates. On the other hand, $\tau_{\mathrm{MCP}}$ or $\tau_{\mathrm{MCP}}^{\#}$ drastically decreased with flexion, and $\tau_{\mathrm{PIP}}$ or $\tau_{\mathrm{PIP}}^{\#}$ was not correlated with the PIP joint angle during cylindrical grip or suspensory hand posture calculated using

MR images of digit III (Fig 6, S5 and S6 Tables). $\sqrt{\frac{\tau_{\mathrm{PIP}}}{\tau_{\mathrm{MCP}}}}$ or $\sqrt{\frac{\tau_{\mathrm{MCP}}^{\#}}{\tau_{\mathrm{PIP}}^{\#}}}$ increased or decreased with

flexion of the finger joint respectively, and $\sqrt{\frac{\tau_{\mathrm{MCP}}^{\#}}{\tau_{\mathrm{PIP}}^{\#}}}$, calculated from MRI of digit III, was significantly larger in *P. hamadryas* than in *Hylobates* spp., not only during a cylindrical grip but also during a suspensory hand posture using GEE, as predicted above in our hypothesis (Figs 5 and 6, S5 and S6 Tables). The analysis of other digits provided similar results (S6 and S7 Figs and S5 and S6 Tables). Similar to the result from our model using bone specimens and CT scan images, these results suggest that the larger torque distribution on the PIP joint and the smaller torque distribution on the MCP joint are more advantageous to support the body weight during brachiation. Alternatively, the traction forces of FDP ($F_{mMCP}$(FDP) and $F_{mMCP}^{\#}$(FDP): indicates the traction force during suspensory hand posture), FDS ($F_{mMCP}$(FDS) and $F_{mMCP}^{\#}$(FDS)), and intrinsic ($F_{mMCP}$(INT) and $F_{mMCP}^{\#}$(INT)) muscles necessary for flexion of the MCP joint were strongly negatively correlated with the MCP joint angle in *Hylobates* spp., *Ateles* sp., and *P. hamadryas* (with the exception of $F_{mMCP}$(INT)), during the adoption of both a suspensory posture and a cylindrical grip (Fig 7, S7 and S8 Tables). This indicates that the efficiency of torque generation by muscle constriction is improved with flexion of the proximal phalanx. $F_{mMCP}^{\#}$(FDP), $F_{mMCP}^{\#}$(FDS), and $F_{mMCP}^{\#}$(INT) were larger in *Hylobates* spp. than in *P. hamadryas* at mild flexion, but they were markedly reduced with an increase in MCP joint angle and became closer to those in *Papio* spp. during a suspensory hand posture (Fig 7, S8 Table). In addition, during both a suspensory hand posture and a cylindrical grip, the traction force of the flexor digitorum muscle necessary for flexion of the PIP ($F_{mPIP}$ and $F_{mPIP}^{\#}$) joint in digit III was negatively correlated with the PIP joint angle in *P. hamadryas* and was smaller in *P. hamadryas* than in *Hylobates* spp. using GEE (Fig 7, S7 and S8 Tables). By contrast, there was no significant correlation between the joint angle and traction force of the FDP muscle necessary for flexion of the DIP joint ($F_{mDIP}$) in any of the species (S7 Table). Similar results in regression analyses and GEE were obtained for the torque and traction force of muscles on the finger joints of digits II and IV (S6 and S7 Figs and S5–S8 Tables). Taking these findings together, the following conclusions are suggested: i) *Papio hamadryas* has a musculoskeletal structure in which the efficiency of torque generation increases with increasing PIP and MCP joint angles, whereas Hylobates spp. only exhibit a functional structure similar to this at the MCP joint; ii) *Papio hamadryas* can generate torque more efficiently at the MCP and PIP joints, but the efficiency of torque generation at the MCP joint in *Hylobates* spp. increases markedly with flexion and approaches that in *Papio* spp. at a greater bending angle, particularly during a suspensory hand posture.

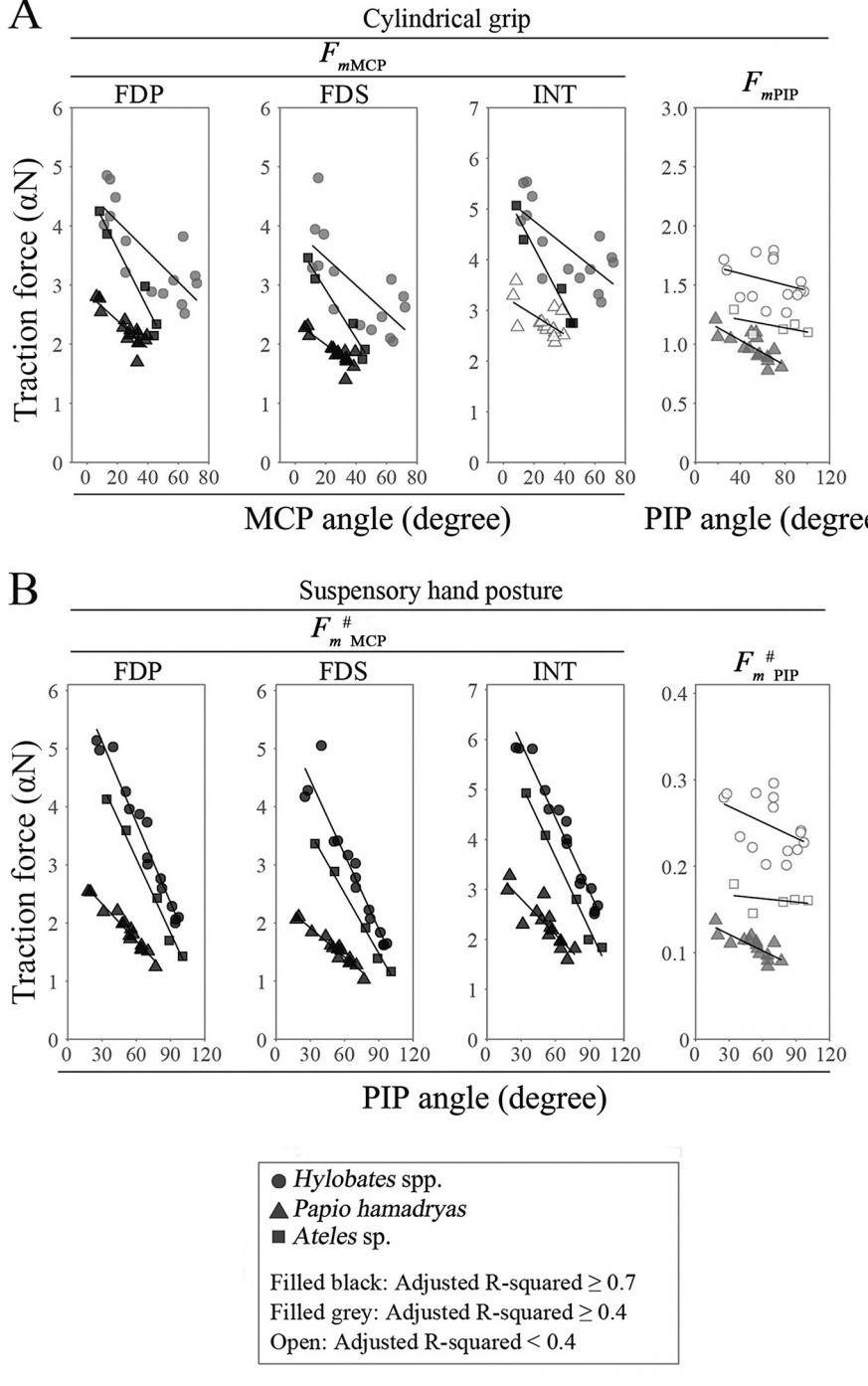

**Fig 7. Correlation between the finger joint angle and the traction force of digital flexor muscles in digit III during a cylindrical grip and suspensory hand posture.** The differences in $F_{mMCP}$, $F_{mPIP}$ during a cylindrical grip (A), $F_{mMCP}^{\#}$, and $F_{mPIP}^{\#}$ during a suspensory hand posture (B) were compared among *Hylobates* spp., *Papio hamadryas*, and *Ateles* sp. using generalized estimating equations (GEE). Circles, triangles, and squares show the data of *Hylobates* spp., *Papio hamadryas*, and *Ateles* sp, respectively. Symbols filled with black, $R^2$ of the regression $\geq$0.7; symbol filled with gray, $R^2$ of the regression $\geq$0.4; open symbol, $R^2$ of the regression <0.4. Regression equations are shown in S7 and S8 Tables.

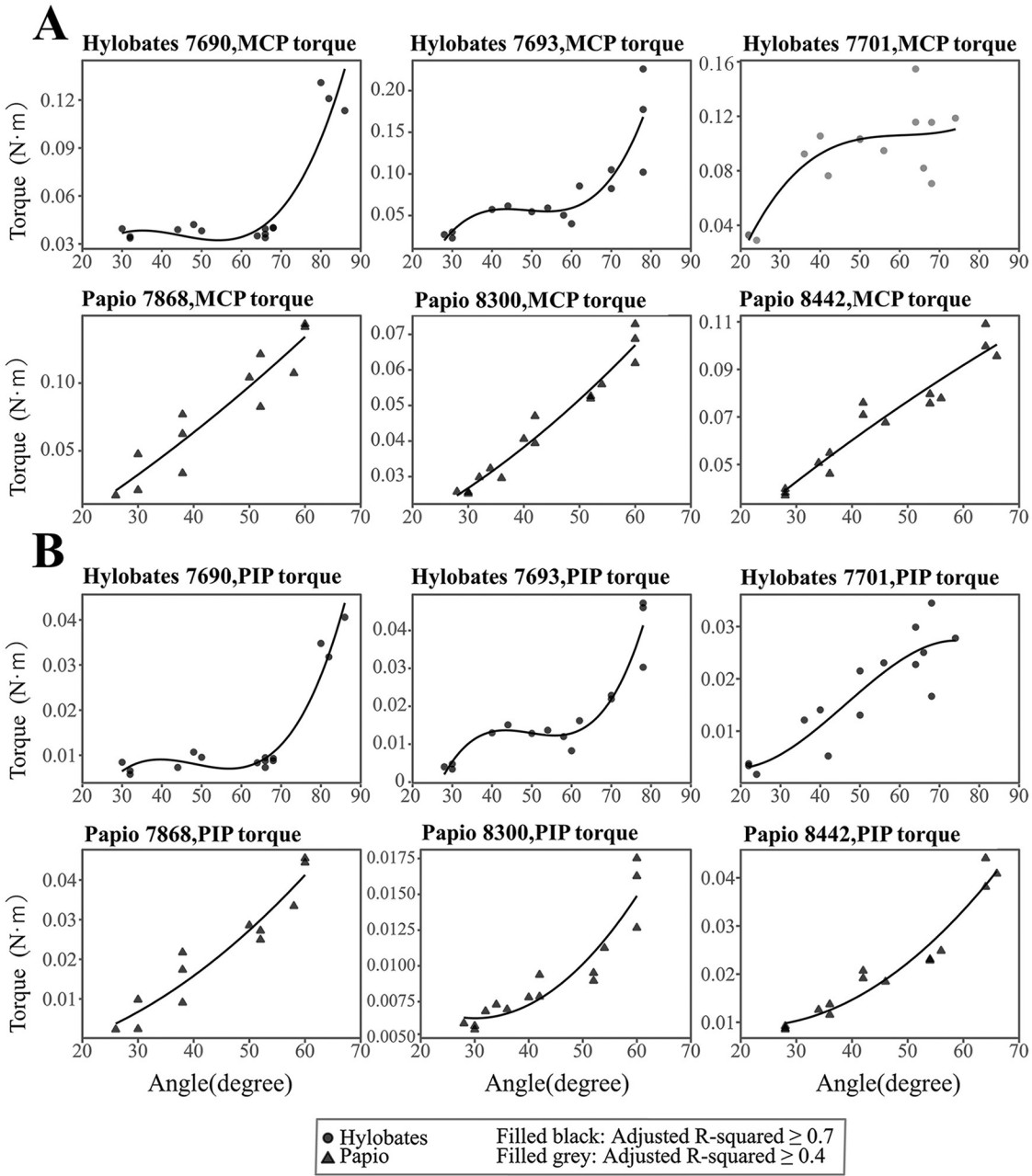

**Fig 8. Correlation between the PIP joint angle and the torque of MCP or PIP joint and between the PIP joint angle and torque ratio in digit III during a suspensory hand posture.** Torque generated on MCP (A) and PIP (B) joints at different PIP joint angles by pulling the digital flexor tendons were plotted in *Hylobates* spp., and *Papio hamadryas*. Symbols filled with black, $R^2$ of the regression ≥0.7; symbol filled with gray, $R^2$ of the regression ≥0.4; open symbol, $R^2$ of the regression <0.4. Regression equations are shown in S9 Table.

### Torque generated by pulling force of the digital flexor muscle tendon during a suspensory hand posture in *Hylobates* spp., and *P. hamadryas* in our mechanical study

There was a strong to moderate and positive correlation between PIP joint angle and the torque on the PIP and MCP joint in *Hylobates* spp. using a cubic polynomial regression (Fig 8A,

S9 Table). The torques on the PIP and MCP joints were stable for less than 68˚ at the PIP angle. These torques strongly and positively correlated with the PIP joint angle in *P. hamadryas* according to a quadratic polynomial regression (Fig 8A, S9 Table); the equation of the torque calculation includes the moment arm as a variable, which was approximated by a quadratic term of the PIP joint angle and a constant term (Fig 6A, Eqs 7 and 9). The torque on the PIP and MCP joints constantly increased with an increase in the PIP joint angle in *P. hamadryas* unlike in *Hylobates* spp. (Fig 8) although the torque could not be compared between *P. hamadryas* and *Hylobates* by GEE because the regression equation was different between these species.

### Torque change upon grasping tree branches of different sizes during quadrupedal locomotion in arboreal and terrestrial quadrupedal primates using our model

Change in torque, which is necessary for supporting the body in a palmigrady posture on a tree, was calculated for tree branches with different diameters. Torques on PIP and MCP joints positively correlated with diameter of the branch grasped by primates. Torque on the MCP joint (r = 0.35 to 1.0) and on the PIP joint (r = 0.8 to 1.0) was greater in the arboreal primates than in the terrestrials (S8 Fig). However, the differences in torque on PIP and MCP joints were small between the arboreal and terrestrial primates when primates grasped a twig (S8 Fig).

## Discussion

### The intrinsic proportions of rays are linked to the locomotor behavior by the cross and triple-ratios, which describe the joint torque distribution specific to a prehensile mode

In this study, we showed that the classifications of primates based on cross-ratio and especially triple-ratio were similar to the classification based on locomotor behavior. The smallest cross and triple-ratios were observed in the arboreal cluster, followed by semi-arboreal and terrestrial clusters, suggesting that the behavioral feature of an arboreal milieu, such as brachiation and suspensory or quadrupedal locomotion on a tree, was linked to small triple and cross-ratios in the hand. Brachiators generally have longer middle and proximal phalanges and a shorter distal phalanx; therefore, the Ph cross-ratio is lower in brachiating primates than in terrestrial primates. The same relationship was obtained for the MPh cross and triple-ratios, which represents a synergy of the proportional information on these two cross-ratios. However, relative to the total length, the proportion of the length of each phalanx or metacarpal bone differed among the species, even within the same cluster (S10 Table), suggesting that the hand can exhibit similar functions when cross and triple-ratios are similar, even if the proportions of these bones are dissimilar among species of the same genus. Whether during the adoption of a suspensory hand posture or quadrupedal locomotion in a tree, the joint torque can be calculated using the same equation; therefore, the similar hand cross and triple-ratios may represent the similar hand properties in regard to the torque balance between suspensory and quadrupedal arboreal primates. In the process in which primates adopt suspensory behavior, one theory insists that the shift from above-branch quadrupedal walking to below-branch walking was attributable to overbalance on the tree branch arising from an increase in the ratio of the body size to support diamete [3, 22] r. *Varecia variegate* and *Lemur catta* walk above and below branches, and there is a shift in the forelimb kinetics for transition to below-

branch walking. Specifically, the forelimb of the primates functions as the primary weight-bearing limb and propulsive organ during below-branch walking, similar to the suspensory hand posture [22]. Meanwhile, our cross-ratio study suggests that a large structural change in the hand is unnecessary for a shift from above-branch locomotion to below-branch locomotion from a viewpoint of the torque balance among the finger joints due to the similar cross-ratio between suspensory species, e.g. *Hylobates* spp. and arboreal quadrupedal species, e.g. *Saimiri sciureus*. In fact, the torques loaded at the MCP and PIP joints for the joint angle, which is influenced by the diameter of branches that primates grasp, were larger in arboreal quadrupedal primates than terrestrial ones. However, the difference in the torque was smaller with a decrease in the diameter of the substrate or an increase in the joint angle (S8 Fig) as shown in the correlation of the torque curve between brachiating and terrestrial primates (Fig 6). These results suggest that small triple and cross-ratios revealed a commonality of the hand structure among arboreal primates in terms of the torque distribution on the joints and distinguished it from that of terrestrial primates. Differences in the patterns of triple and cross-ratios from digits II to V between *H. sapiens* and terrestrial quadrupedal primates may be due to differences in the functions of hands, which are used for grasping/object manipulation and running in quadrupedal primates or just for performing the former action exclusively. Thus, cross and triple-ratios may represent the optimized intrinsic proportion of rays for different prehension modes, such as precision or a power grasp including a cylindrical grasp and brachiation, or quadruped locomotion on the ground or in trees.

## Torque generation efficiency and property on the PIP and MCP joints changing with their joint angles are optimized to the prehensile mode

**Suspensory hand posture.** By calculating the torque ratio, which is a component of equations for cross and triple-ratios, it was shown that arboreal primates have a hand structure in which the torque load is reduced on the MCP joint during the adoption of a suspensory hand posture. In addition, the efficiency of torque generation on the MCP joint calculated from MRI data of brachiating primates increased markedly upon flexion via torque reduction because of the characteristic intrinsic proportion of rays, while lengthening of the moment arm also contributed to improving the torque generation efficiency in terrestrial quadrupedal primates. In the maximum pulling task with a hook grip similar to the suspensory hand posture, the contribution of the phalanx force in *H. sapiens* was reported to be largest in the proximal phalanx (45%), followed by the middle (34%) and distal (21%) phalanges [23, 24]. These reports suggest that a large moment is generated on the MCP joint followed by the PIP joint when a suspensory hand posture is adopted. To overcome a large torque load on the MCP joint, improvement in the torque generation efficiency with flexion may be beneficial for hanging from tree branches. Meanwhile, the torques on the PIP and MCP joints were stable in *Hylobates* spp. during flexion of the PIP joint less than 68˚ in our mechanical study. Stable torque may stem from the stable moment arm of the flexor muscle on the PIP and MCP joints at different PIP joint angles (See Eq 8); therefore, our model is consistent with the torque generation mode measured in our mechanical study. An increase in the torque exerted on the joints, which results from an increase in the force exerted on the phalanges, with flexion may be discouraged in brachiating primates. This is because a large force produces a large friction between the finger and a tree branch, which may interfere with swinging. Nevertheless, the torques exerted on the MCP and PIP joints gradually increased with flexion of the PIP joint due to the lengthening moment arm in *P. hamadryas* even if the traction power of the flexor tendons is constant. Taken together, these results suggest that the

hand morphology in *Hylobates* spp. has evolved to be suitable for brachiation, whereas that in *P. hamadryas* has evolved to be suitable for a firm grip on a cylinder rather than a suspensory hand posture.

**Cylindrical grip and terrestrial quadrupedal locomotion.** By contrast, the force contribution was smallest in the metacarpal bone, followed by the proximal, middle, and distal phalanges in a maximum isometric grip, unlike a hook grip when *H. sapiens* grasped a cylindrical handle measuring 35–50 mm in diameter with the thumb and opposite fingers [25]. The proportions of force exerted by the distal, middle, and proximal phalanges in *H. sapiens* were found to be 35.9%, 32.4%, and 31.7%, respectively, in the gripping task [26]. These reports suggest that the torque generated on the PIP joint should be greater in the gripping task (power grip) than in the pulling task (hook grip). From this perspective, the hand structure of terrestrial primates may have evolved to be optimized for a power grasp using digits I to V and the palm, that is, to increase torque generation efficiency with flexion of the middle phalanx, as shown in our study. Meanwhile, the olive baboon (*P. anubis*) used a digitigrade hand posture at slower speeds with lower ground reaction force (GRF) during terrestrial locomotion [27]. The hand posture transitioned to palmigrade at higher speeds with a higher GRF, in which the proximal portion of the metacarpal bone makes contact with the ground [27]. In addition, the peak pressure in the finger was notably higher in *Papio* spp. at higher speeds than at lower ones during terrestrial locomotion [27]; thus, the MCP joint moment may be quite large at higher speeds. Therefore, a reduced torque distribution and high torque generation efficiency at extension and mild flexion of the fingers may be advantageous for terrestrial quadruped species, as shown in our study. Kikuchi [28] reported that *P. hamadryas* has a relatively large physiological cross-sectional area (PCSA), but the Hylobatidae have a small PCSA in their digital flexor muscles. This study also suggested that heavier primates have a large PCSA because they need more muscle force to move, climb, and lift their own bodies [28]. Hence, great muscular strength may also contribute to large torque generation on the PIP and MCP joints for a cylindrical grip and quadrupedal locomotion in *Papio* spp. However, the hand skeletal structure linked with cross and triple-ratios, together with the moment arm of the flexor muscles, may be important for primates to exert a peculiar behavioral function from the perspective of torque generation efficiency during a power grasp.

## Conclusions

Our study revealed that the cross or triple-ratios in primates, which parameterizes the intrinsic proportions of rays, can provide information about the optimal structure required for exerting well-balanced torque on the DIP, PIP, MCP, and CMC joints for various prehensile modes. Therefore, primates could be classified into the group consistent with their behavior using the cross- and triple-ratios. Our study suggests that the torque is allocated less in the MCP joint, especially during flexed posture of the fingers, i.e., a suspensory hand posture, in the arboreal primates, but especially during extended position, i.e., quadrupedal locomotion, in the terrestrial primates. Moreover, the torque exerted on MCP and PIP joints were stable under the isotonic muscle contraction of digital flexors during a suspensory hand posture in suspensory primates at moderate degree of finger flexion. However, in the terrestrial primates, the torque gradually increased with an increase of the joint angles so that the torque can be sufficiently generated in extended and flexed hand positions. We deduced that stable and sufficient, but not exceedingly large torque generation profiles on the MCP and PIP joints during moderate flexion of fingers in brachiating primates is advantageous for brachiation. This is because the frictional force, which interferes with the swing motion of the forelimb, is low between the fingers and a tree branch. In contrast, the terrestrial primates can obtain greater propulsive force

in quadrupedal locomotion or squeezing power in firm grip postures because suitable power can be obtained in any degree of flexion of the fingers.

Thus, our study provides evidence that the intrinsic proportion of rays is mechanically optimized for varied primate behaviors. Furthermore, the cross-ratio results provide mechanical information that could be useful for the creation of artificial hands or for understanding prehensile function or locomotion.

## Supporting information

**S1 Fig. Three-dimensional reconstruction images of the phalanges, metacarpal bones, and flexor tendons of digit III.** In *Hylobates* spp. (A, B), *Ateles* sp. (C, D), and *Papio hamadryas* (E, F), the phalanges, metacarpal bones, and flexor digitorum tendons were reconstructed from MR images of digit III. Their lateral views at extension (A, C, E) and flexion (B, D, F) are shown. The positional relationship between bones and tendons was changed by the finger posture.
(TIF)

**S2 Fig. Relationship between the joint torque and bone length in a simple joint model.** The holding torque is proportional to the square of the length of the proximal phalanx. *b*, length of the proximal phalanx; *f*, reaction force (thick arrows) from the central axis of the cylinder to the bone; *r*, radius of the cylinder; $\theta$, the angle between *f* and *x*-axis; $\tau_s$, joint torque.
(TIF)

**S3 Fig. Calculation of joint torques in the suspensory hand posture.** (A) The proximal interphalangeal (PIP) joint is positioned on the top of the support during a suspensory hand posture. The center and radius of the cylinder and torques loaded on metacarpophalangeal (MCP) and proximal interphalangeal (PIP) joints are defined as $O$, $r$, $\tau_{\mathrm{MCP}}$, and $\tau_{\mathrm{PIP}}$, respectively. $\vec{f}$, the reaction force against the gravity during the suspensory hand posture; $l_{\mathrm{pp}}$, $l_{\mathrm{ip}}$, and $l_{\mathrm{dp}}$, the lengths of the proximal, middle, and distal phalanges; $\beta$, $\gamma$, and $\delta$, inscribe angles of $l_{\mathrm{pp}}$, $l_{\mathrm{ip}}$, and $l_{\mathrm{dp}}$. (B) The arc of the cylinder is defined as $\hat{l}$, and the chord and center angle of the arc $\hat{l}$ are defined as $\vec{l}$ and $\theta$. Holding torques of MCP and PIP during brachiation with or without using the distal phalanx are calculated by the integral of the infinitesimal torques ($d\tau$), which is a cross product of $\vec{l}$ and $\vec{f}$.
(TIF)

**S4 Fig. Relationship between torque ratio and phalangeal lengths.** The center and radius of the cylinder are defined as $O$, and $r$, respectively. $\theta_3$, proximal interphalangeal (PIP) joint angle; $l_{\mathrm{pp}}$, $l_{\mathrm{ip}}$, and $l_{\mathrm{dp}}$, the lengths of the proximal, middle, and distal phalanges; $\beta$, $\gamma$, and $\delta$, inscribe angles of $l_{\mathrm{pp}}$, $l_{\mathrm{ip}}$, and $l_{\mathrm{dp}}$; $L$, the line, which is parallel with the horizontal line (*x*-axis) and passes through the center of the proximal interphalangeal joint.
(TIF)

**S5 Fig. Normalized moment arms in digits II, IV, and V.** Normalized moment arm, $\frac{l_m}{L}$, was calculated on distal interphalangeal (DIP), proximal interphalangeal (PIP), and metacarpophalangeal (MCP) joints. Regression equations are shown in S3 Table.
(TIF)

**S6 Fig. Correlation between the finger joint angle and torque, torque ratio, or traction force of digital flexor muscles calculated using MR images of digits II, IV, and V during cylindrical grip.** The differences in $\sqrt{\frac{\tau_{\mathrm{PIP}}}{\tau_{\mathrm{MCP}}}}$ (A), $\tau_{\mathrm{MCP}}$ (B) and $\tau_{\mathrm{PIP}}$ (C), $F_{mMCP}$ of FDP (D),

FDS (E) or intrinsic muscles (F), and $F_{m\text{PIP}}$ (G) were compared among *Hylobates* spp., *Papio hamadryas*, and *Ateles* sp. using generalized estimating equations (GEE). Circles, triangles, and squares show the data of *Hylobates* spp., *Papio hamadryas*, and *Ateles* sp., respectively. Symbols filled with black, $R^2$ of the regression $\geq$0.7; symbol filled with gray, $R^2$ of the regression $\geq$0.4; open symbol, $R^2$ of the regression <0.4. Regression equations are shown in S4 and S6 Tables.
(TIF)

**S7 Fig. Correlation between the finger joint angle and torque, torque ratio, or traction force of digital flexor muscles calculated using MR images of digits II, IV, and V during a suspensory hand posture.** The differences in $\sqrt{\frac{\tau_{\text{MCP}}^{\#}}{\tau_{\text{PIP}}^{\#}}}$ (A), $\tau_{\text{MCP}}^{\#}$ (B) $\tau_{\text{PIP}}^{\#}$, (C), $F_{m\text{MCP}}^{\#}$ of FDP (D), FDS (E), or intrinsic muscles (F), and $F_{m\text{PIP}}^{\#}$ (G) were compared among *Hylobates* spp., *Papio hamadryas*, and *Ateles* sp. using generalized estimating equations (GEE). Circles, triangles, and squares show the data of *Hylobates* spp., *Papio hamadryas*, and *Ateles* sp., respectively. Symbols filled with black, $R^2$ of the regression $\geq$0.7; symbol filled with gray, $R^2$ of the regression $\geq$0.4; open symbol, $R^2$ of the regression <0.4. Regression equations are shown in S6 and S7 Tables.
(TIF)

**S8 Fig. Calculation of joint torques during the quadrupedal locomotion on a tree.** (A) A midpoint of a metacarpal bone is positioned on the top of the support. The center and radius of the cylinder and torques loaded on metacarpophalangeal (MCP) and proximal interphalangeal (PIP) joints are defined as $O$, $r$, $\tau_{\text{MCP}}$, and $\tau_{\text{PIP}}$, respectively. $\vec{f}$, the reaction force against the gravity during the arboreal quadrupedal locomotion; $\alpha$, $\beta$, $\gamma$, and $\delta$, inscribe angles of the lengths of the metacarpal bone, and proximal, middle, and distal phalanges. The torques on (B) MCP and (C) PIP joints, regression equations, and determinant coefficients ($R^2$) were shown in arboreal (open circle) and terrestrial (open triangle) quadrupedal primates.
(TIF)

**S1 Appendix. Information about non-human primate experiment.**
(DOCX)

**S2 Appendix. Torque calculation on the joints during a power grasp.**
(DOCX)

**S3 Appendix. Torque calculation on the joints during suspensory hand postures.**
(DOCX)

**S4 Appendix. Relationship between the cross-ratio and torque ratios between the joints.**
(DOCX)

**S5 Appendix. Calculation of torque on the joints during quadrupedal locomotion on a tree branch.**
(DOCX)

**S1 Table. The information of the primates examined in this study.**
(DOCX)

**S2 Table. Abbreviation table.**
(DOCX)

**S3 Table. Parameters of MRI sequences.**
(DOCX)

**S4 Table. Regression equations of the normalized moment arm on the finger joint angle during a cylindrical grip.**
(DOCX)

**S5 Table. Regression equations of the torque and torque ratio on the finger joint angle during a cylindrical grip.**
(DOCX)

**S6 Table. Regression equations of the torque and torque ratio on the finger joint angle during a suspensory hand posture.**
(DOCX)

**S7 Table. Regression equations of the traction force of the flexor tendons on the finger joint angle during a cylindrical grip.**
(DOCX)

**S8 Table. Regression equations of the traction force of the flexor tendon on the PIP joint angle during a suspensory hand posture.**
(DOCX)

**S9 Table. Regression equations of the corrected torque on the PIP joint angle during a suspensory hand posture.**
(DOCX)

**S10 Table. Proportion of the length of finger bones, and triple- and cross-ratios.**
(DOCX)

## Acknowledgments

We are grateful to the Natural History Museum in Gothenburg, Natural History Museum of Denmark in Copenhagen, Dr. Shin-ichiro Kawada (National Museum of Nature and Science in Tokyo), and the Primate Research Institute Kyoto University. We also thank Dr. Toshiko Tsumori (Department of Nursing, Faculty of Health and Welfare, Prefectural University of Hiroshima) and Dr. Yukihiko Yasui (Department of Anatomy and Morphological Neuroscience, Faculty of Medicine, Shimane University) for giving us access to several monkey skeletons.

## Author Contributions

**Conceptualization:** Toshihiro Tamagawa, Torbjörn Lundh, Kenji Shigetoshi, Jun Udagawa.

**Data curation:** Toshihiro Tamagawa, Torbjörn Lundh, Kenji Shigetoshi, Norihisa Nitta, Noritoshi Ushio, Toshiro Inubushi, Akihiko Shiino, Takayuki Inoue, Kodai Hino, Shigehiro Morikawa, Tomoko Kimura, Jun Udagawa.

**Formal analysis:** Toshihiro Tamagawa, Torbjörn Lundh, Yutaka Mera, Shuji Sawajiri, Yasuhiro Uchimura, Jun Udagawa.

**Funding acquisition:** Jun Udagawa.

**Investigation:** Toshihiro Tamagawa, Norihisa Nitta, Noritoshi Ushio, Toshiro Inubushi, Akihiko Shiino, Takayuki Inoue, Kodai Hino, Shigehiro Morikawa, Shigeyuki Naka, Satoru Honma, Tomoko Kimura, Yasuhiro Uchimura, Jun Udagawa.

**Methodology:** Toshihiro Tamagawa, Torbjörn Lundh, Kenji Shigetoshi, Toshiro Inubushi, Akihiko Shiino, Anders Karlsson, Yutaka Mera, Kodai Hino, Masaru Komori, Shuji Sawajiri, Shigeyuki Naka, Satoru Honma, Shinji Imai, Naoko Egi, Hiroki Otani, Jun Udagawa.

**Project administration:** Jun Udagawa.

**Supervision:** Jun Udagawa.

**Validation:** Jun Udagawa.

**Writing – original draft:** Toshihiro Tamagawa, Torbjörn Lundh, Anders Karlsson, Jun Udagawa.

**Writing – review & editing:** Toshihiro Tamagawa, Torbjörn Lundh, Kenji Shigetoshi, Anders Karlsson, Yutaka Mera, Naoko Egi, Hiroki Otani, Jun Udagawa.

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
