## [Decision Letter · Decision Letter 0]

7 Jan 2020

PONE-D-19-21332

Correlation between musculoskeletal structure of the hand and primate locomotion: Morphometric and mechanical analysis in prehension using the cross- and triple-ratios

PLOS ONE

Dear Dr. Udagawa,

Thank you for submitting your manuscript to PLOS ONE. After careful consideration, we feel that it has merit but does not fully meet PLOS ONE’s publication criteria as it currently stands. Therefore, we invite you to submit a revised version of the manuscript that addresses the points raised during the review process.  As you make revisions, please also pay particular attention to improving the clarity of your presentation and reducing the "density" of the information conveyed.

We would appreciate receiving your revised manuscript by Feb 21 2020 11:59PM. To enhance the reproducibility of your results, we recommend that if applicable you deposit your laboratory protocols in protocols.io, where a protocol can be assigned its own identifier (DOI) such that it can be cited independently in the future. For instructions see: http://journals.plos.org/plosone/s/submission-guidelines#loc-laboratory-protocols

We look forward to receiving your revised manuscript.

Kind regards,

Xudong Zhang

Academic Editor

PLOS ONE

Journal Requirements:

2. We note that your study involved measurements performed on non-human primates, specifically the crab-eating macaques. In order to comply with PLOS ONE's guidelines for non-human primate experiments (http://journals.plos.org/plosone/s/submission-guidelines#loc-non-human-primates), please provide additional details regarding housing conditions, feeding regimens, environmental enrichment, and all relevant steps taken to alleviate suffering (anesthesia, analgesia, details about humane endpoints, euthanasia, etc.). Also indicate how often animal care staff monitored the health and well-being of the animals and the criteria used to make such assessments. Lastly, specify the disposition of animals at the end of the study (e.g. euthanasia, returned to home colony, etc.). If animals were euthanized following the study, please provide the method of sacrifice.

Reviewers' comments:

Reviewer's Responses to Questions

**Comments to the Author**

1. Is the manuscript technically sound, and do the data support the conclusions?

Reviewer #1: Yes

Reviewer #2: Yes

2. Has the statistical analysis been performed appropriately and rigorously? 

Reviewer #1: I Don't Know

Reviewer #2: No

3. Have the authors made all data underlying the findings in their manuscript fully available?

Reviewer #1: No

Reviewer #2: No

4. Is the manuscript presented in an intelligible fashion and written in standard English?

Reviewer #1: Yes

Reviewer #2: Yes

5. Review Comments to the Author

Reviewer #1: Review of PONE-D-19-21332

Udagawa et al. “Correlation between musculoskeletal structure of the hand and primate locomotion:

Morphometric and mechanical analysis in prehension using the cross- and triple-ratios.”

Summary: Udagawa and colleagues present the result of an analysis of digital proportions using cross and triple ratios to examine differences among primates of varying locomotor behaviors. They also conduct in vitro experimental work (using primate cadavers) to tie these proportions to torque distributions on the joints of the fingers during grasping. They find that when the intrinsic digital proportions are quantified using these ratios, they cluster primates (approximately) into three locomotor groups: arboreal, semiarboreal, and terrestrial. Moreover, they show that these ratios do correspond reasonably well with quantified torque distributions from their cadaver experiments and indicate torque production capabilities that should be beneficial in the habitual hand postures used by the primates they measured.

Evaluation: Overall, this paper presents an interesting study with data relevant to understanding the form and function of the primate hand. The approach of coupling morphometric data with a noninvasive method of quantifying torque in the cadaver hands is effective in looking at the basic correlation between structure and function. However, I have the following concerns and questions that I would like to see addressed.

1. The paper is quite difficult to read and follow. Part of this is because of the dense biomechanics, which always makes for a more challenging read. However, the hypotheses and the predictions that follow from those hypotheses could be more clearly and simply stated. The authors should then return to these hypotheses point by point at the end of the paper. I believe this simple change in the format of the paper would help them turn what might be a somewhat obscure paper for specialists into a paper with wider impact.

2. As with the hypotheses, what statistical procedures were applied to what aspects of the study and what samples was difficult to understand. For example (from page 21): “The significance of differences in the cross and triple-ratios and the torque ratio between H. sapiens and other primates were assessed using Student’s t-test, and intraspecific differences were analyzed using Scheffe’s post hoc test in SPSS.”

Were humans treated differently and compared with all of the other taxa? What do they mean by “intraspecific differences.” Or was that a typo and they mean interspecific differences here?

3. Box-and-whisker plots of the cross and triple ratios for each taxon would be helpful.

4. Overall, the existing literature on primate hand postures and hand use has not been well cited. The authors use a fairly simplistic locomotor classification of arboreal versus semiarboreal versus terrestrial, but there is no indication of how these categories have been assigned. What data are they using? Further, there is some variation among these primates in how they habitually apply their hands to the ground or other substrate: palmigrade versus digitigrade versus knuckle-walking. Those distinctions are largely ignored in this paper. Other terms are not used very precisely (e.g., most primatologists consider gibbons to be the only true “brachiators”—that term have a specific meaning applying to the ricochetal form of arm swinging).

5. I don’t understand how the joint angles were varied in their cadaver experiments. Was that accomplished by their varying the size of the substrate that was grasped? Or was that a separate process? The authors say “five flex postures,” but it’s not clear exactly what they mean or how they did it. That needs better a better explanation given its an important part of the study.

6. Can the authors comment on how real primate body masses might affect the outcome? The cadavers are all treated to the same loading regime, but these primates vary in their body mass, so in vivo conditions would be quite different.

7. How does phalangeal shaft curvature factor into the joint torque calculations? These primates all vary considerably in the degree of phalangeal curvature.

Reviewer #2: Thank you for the opportunity to review this manuscript, “Correlation between musculoskeletal structure of the hand and primate locomotion: Morphometric and mechanical analysis in prehension using the cross- and triple-ratios”. This study uses novel morphometric indices and innovating in vitro modelling to test the hypothesis that primate hands may be “built” to most efficiently produce torque in primary locomotor mode (i.e., terrestrial quadrupedalism, arboreal quadrupedalism, or below branch suspension). The authors find broad support for this prediction.

In general, I found study very impressive, but also very dense. I fear that the import of the research may be lost among the stream of equations, abbreviations, and results that pour forth. My primary recommendation is that the authors revise the manuscript in ways that will make the argument easier to follow for potential readers. I have some recommendations on this below. I use the author-supplied line numbers when referencing specific sections of the text.

(Line 82) Sapajus apella NOT Sapajus paella (common spell check casualty…)

(Lines 96-110) The authors clearly have hypotheses and predictions concerning both digit ratios and torque distributions in the different locomotor groups. They should state these hypotheses here, rather than frame the research in terms of questions.

(Methods, generally) There are many separate metrics and variables considered in this study. A table of all abbreviations would greatly help the reader remember what symbols are used for what metric or variable. Also, where possible, I’d encourage the authors to use variable names or abbreviations that are meaningful to the reader, rather than arbitrary letters or numbers. For instance, instead of a, b, c, and d for metacarpal and proximal, intermediate, and distal phalangeal lengths, why not l[mc], l[pp], l[ip], and l[dp]? Similarly, instead of tau[1], tau[2], etc for the holding torques at the joints, why not tau[MCP] etc? Such a convention would relieve the reader from having to remember what each symbol stands for.

(Table 1) I don’t feel it’s appropriate to group humans with other terrestrial primates, given that humans don’t use their hands for locomotion. The authors seem to draw a similar conclusion much later in the paper (lines 652-655).

(Line 328) In their review for force platform and kinematic analysis, Biewener and Full (1992) note that calibration curves for force transducers should typically be ≥ 0.99. I recommend fitting a quadratic regression curve to these calibrations to improve fit and force calculation.

(Line 337) hamadryas NOT hamadryases

(Statistics, generally) Given that the authors are comparing 25 species, some correction should be made for phylogenetic relatedness. Species cannot be considered independent data points and doing so can conflate findings that reflect phylogeny with those thought to reflect locomotion (i.e., all terrestrial primates are catarrhines). See Barr and Scott (2014) for a review of phylogenetic correction in the context of DFA.

(Figure 3) Use species abbreviations instead of arbitrary letter abbreviations for species (i.e., Ss for Saimiri sciureus, Pp for Pongo pygmaeus, etc).

(Table 2) The Percent of variance and Cumulative % rows show the same information (i.e., both are cumulative).

(Line 457) Scheffe’s NOT Sheffe’s

(Figure 8) Please label the axes for each panel figure.

(Discussion, generally) Following my above recommendation to end the introduction section with hypotheses and predictions, I recommend returning to hypotheses and predictions in the discussion and specifically reviewing if these predictions were met.

(Line 705) Patel and Wunderlich should be cited in the requisite format (i.e., numerically)

(Data availability) Raw data have not been made available, as per PLoS’s policy.

REFERENCES CITED

Barr WA, and Scott RS. 2014. Phylogenetic comparative methods complement discriminant function analysis in ecomorphology. Am J Phys Anthropol 153(4):663-674.

Biewener AA, and Full RJ. 1992. Force platform and kinematic analysis. In: Biewener AA, editor. Biomechanics: Structures and Systems. Oxford: Oxford University Press. p 45-73.

6. PLOS authors have the option to publish the peer review history of their article (what does this mean?). If published, this will include your full peer review and any attached files.

Reviewer #1: No

Reviewer #2: Yes: Jesse Wyatt Young

---

## [Author Response · Author response to Decision Letter 0]

21 Feb 2020

Dear Prof. Xudong Zhang,

Thank you for reviewing our manuscript (PONE-D-19-21332) titled “Correlation between musculoskeletal structure of the hand and primate locomotion: Morphometric and mechanical analysis in prehension using the cross- and triple-ratios.”. We appreciate the insightful comments provided by the editor and reviewers. We have revised our manuscript in accordance with the reviewers’ suggestions. Furthermore, we have added the information about non-human primate experiments in S1 Appendix. The changed portions (marked in yellow highlights in the revised manuscript) and our responses to the reviewers’ comments are enclosed.

This manuscript has carefully been reviewed by an experienced editor who specializes in editing papers written by scientists whose native language is not English.

We hope that you find our manuscript suitable for publication and look forward to hearing from you at your earliest convenience.

Yours sincerely,

Jun Udagawa

Department of Anatomy

Shiga University of Medical Science

Review Comments to the Author

Reviewer #1: Review of PONE-D-19-21332

Evaluation: Overall, this paper presents an interesting study with data

relevant to understanding the form and function of the primate hand. The

approach of coupling morphometric data with a noninvasive method of

quantifying torque in the cadaver hands is effective in looking at the

basic correlation between structure and function. However, I have the

following concerns and questions that I would like to see addressed.

1. The paper is quite difficult to read and follow. Part of this is

because of the dense biomechanics, which always makes for a more

challenging read. However, the hypotheses and the predictions that

follow from those hypotheses could be more clearly and simply stated.

The authors should then return to these hypotheses point by point at the

end of the paper. I believe this simple change in the format of the

paper would help them turn what might be a somewhat obscure paper for

specialists into a paper with wider impact.

Response: Thank you for your suggestion. We have now stated our hypotheses and predictions in the Introduction (Lines 96-119), and have discussed the outcomes in regard to these predictions in the Conclusion (Lines 784-805).

2. As with the hypotheses, what statistical procedures were applied to

what aspects of the study and what samples was difficult to understand.

For example (from page 21): “The significance of differences in the

cross and triple-ratios and the torque ratio between H. sapiens and

other primates were assessed using Student’s t-test, and intraspecific

differences were analyzed using Scheffe’s post hoc test in SPSS.”

Were humans treated differently and compared with all of the other taxa?

What do they mean by “intraspecific differences.” Or was that a typo and

they mean interspecific differences here?

Response: Thank you for your suggestion. I have specified the statistical procedures that were applied to each experiment (Lines 421-431).

3. Box-and-whisker plots of the cross and triple ratios for each taxon

would be helpful.

Response : Thank you for your suggestion. I have changed Fig. 4.

4. Overall, the existing literature on primate hand postures and hand

use has not been well cited. The authors use a fairly simplistic

locomotor classification of arboreal versus semiarboreal versus

terrestrial, but there is no indication of how these categories have

been assigned. What data are they using? Further, there is some

variation among these primates in how they habitually apply their hands

to the ground or other substrate: palmigrade versus digitigrade versus

knuckle-walking. Those distinctions are largely ignored in this paper.

Other terms are not used very precisely (e.g., most primatologists

consider gibbons to be the only true “brachiators”—that term have a

specific meaning applying to the ricochetal form of arm swinging).

Response cation of primates in this study. We classified arboreal (suspensory), arboreal (quadrupedal), semi-arboreal, and terrestrial primates based on the literature cited in the manuscript [i-vi]. Furthermore, we revised the manuscript to classify gibbons, spider monkeys, and orangutan as “suspensory species” instead of using the term “brachiating primates.” We have added details of the classification in our manuscript (Lines 138-155). The reviewer has raised an important point regarding variations among how primates habitually apply their hands to the ground or other substrates. Arboreal primates use palmigrade postures to grip trees (Lines 405-406). Terrestrial primates typically use a digitigrade hand posture during quadrupedal locomotion on the ground, but some (e.g., baboons) also use a palmigrade hand posture at higher speeds (Lines 766-768). For both the digitigrade and palmigrade hand postures, we use our cylindrical grip model because torque direction on the joint is the same between cylindrical grip and a digitigrade or palmigrade hand posture. We have cited another source regarding a digitigrade and palmigrade hand posture in primates [vii]. Pan troglodytes and Gorilla gorilla travel on the ground by knuckle-walking, as the reviewer pointed it out. However, the torque on the PIP joint during knuckle-walking through the contraction of the extensor digitalis muscles is generated in the opposite direction when compared to the torque generated during power grasping. Hence, joint torque was not considered in knuckle-walking in this study as we could not accurately calculate the torque during knuckle-walking using our model (Lines 246-250). We have focused on the suspensory hand posture and the cylindrical grip model, which can be applied to a palmigrade hand posture on the tree and the ground in this manuscript.

i. Fleagle JG. Primate Adaptation and Evolution: Elsevier Science; 2013.

ii. McGraw WS. Positional Behavior of Cercopithecus petaurista. International Journal of Primatology. 2000;21(1):157-82. doi: 10.1023/a:1005483815514.

iii. Mcgraw WS, Zuberbühler K. Socioecology, predation, and cognition in a community of West African monkeys. Evolutionary Anthropology: Issues, News, and Reviews. 2008;17(6):254-66. doi: doi:10.1002/evan.20179.

iv. Schmitt D. Forelimb Mechanics during Arboreal and Terrestrial Quadrupedalism in Old World Monkeys. In: Strasser E, Fleagle JG, Rosenberger AL, McHenry HM, editors. Primate Locomotion: Recent Advances. Boston, MA: Springer US; 1998. p. 175-200.

v. Crompton RH, Sellers WI, Thorpe SK. Arboreality, terrestriality and bipedalism. Philos Trans R Soc Lond B Biol Sci. 2010;365(1556):3301-14. doi: 10.1098/rstb.2010.0035. PubMed PMID: 20855304; PubMed Central PMCID: PMCPMC2981953.

vi. Hunt KD. Positional Behavior in the Hominoidea. Int J Primatol. 1991;12(2):95-118. doi: 10.1007/BF02547576.

vii. Patel BA. Functional morphology of cercopithecoid primate metacarpals. J Hum Evol. 2010;58(4):320-37. Epub 2010/03/12. doi: 10.1016/j.jhevol.2010.01.001. PubMed PMID: 20226498.

5. I don’t understand how the joint angles were varied in their cadaver

experiments. Was that accomplished by their varying the size of the

substrate that was grasped? Or was that a separate process? The authors

say “five flex postures,” but it’s not clear exactly what they mean or

how they did it. That needs better a better explanation given its an

important part of the study.

Response: We appreciate the reviewer’s comment on this point. We added the following: The hand was scanned in the extended position and in the flexed position while grasping wooden clubs measuring 2 and 3 cm in diameter. For Hylobates spp. and Ateles sp., the hand was also scanned in the flexed position while tightly or mildly grasping a wooden club measuring 4 cm in diameter. We obtained scans of five different flex positions, with five different DIP, PIP, and MCP angles, to improve the accuracy of the regression curve of the torque and the moment arm. Similarly, the hand of P. hamadryas was scanned in the extended position and the flexed position while grasping wooden clubs measuring 2 and 3 cm in diameter, in addition to the flexed position while tightly or mildly grasping a wooden club measuring 1.5 cm in diameter (Lines 299-308).

6. Can the authors comment on how real primate body masses might affect

the outcome? The cadavers are all treated to the same loading regime,

but these primates vary in their body mass, so in vivo conditions would

be quite different.

Response: Thank you very much for your suggestion. In accordance with the reviewer’s comment, I have added the explanation on the impact of body mass on our model (Lines 394-400). We compared the torque-PIP joint angle curve profiles, but not the magnitude of the torque, to examine differences in torque output properties between Hylobates and Papio. This allowed us to compare the functional features of the hand independently. The torque-PIP joint angle curve profile represents torque output properties with respect to different degrees of finger flexion. The curve profile, but not the absolute value of the torque, may be independent of individual body mass during a suspensory hand posture because a magnitude of fH2 or fH3 in Eq. 10 is proportional to the body mass. Hence, torque-PIP joint angle curve profiles were compared between Hylobates and Papio to examine the difference in torque output properties on the MCP and PIP joints between a suspensory species and a terrestrial quadrupedal primate.

7. How does phalangeal shaft curvature factor into the joint torque

calculations? These primates all vary considerably in the degree of

phalangeal curvature.

Response: We appreciate the reviewer’s comment on this point. We agree with the reviewer that the phalangeal shaft curvature affects the angle between the direction tangential to the shaft curve and the direction of the force. However, in our model, the effect of the phalangeal curvature on the torque did not need to be considered because of the following reasons:

In our simple model, we assumed that the finger surface fits the surface of a cylinder since there are skin and subcutaneous tissue between the phalangeal bone and a cylinder. Based on this assumption, the reaction force is normal to the finger surface, and the joint torque is described in the manuscript:

(b^2 |f|)/2r

where, b is the length of the phalanx, r is the radius of the object grasped, and f is the reaction force exerted on the phalanx. Furthermore, the torque is a cross product of the reaction force exerted on the phalanx and the distance from the joint, but not the shaft curve length. Therefore, the joint torque is linked to the phalangeal length, but is not independent of the phalangeal shaft curvature. We have changed the following sentence, “we can assume that the reaction force from the object grasped per unit area is constant on the bone surface”, to “… is constant on the finger surface” (Line 232). In addition, we have changed “the metacarpal bone” into “the metacarpal region” (Line 966) according to the same reasoning described above.

Similarly, the joint torque was calculated under the same assumption in our study. Therefore, the torque was affected by the distance between the joints, which is related to an inscribe angle, however, it was not affected by the phalangeal shaft curvature.

Reviewer #2:

In general, I found study very impressive, but also very dense. I fear

that the import of the research may be lost among the stream of

equations, abbreviations, and results that pour forth. My primary

recommendation is that the authors revise the manuscript in ways that

will make the argument easier to follow for potential readers. I have

some recommendations on this below. I use the author-supplied line

numbers when referencing specific sections of the text.

(Line 82) Sapajus apella NOT Sapajus paella (common spell check casualty

…)

Response: We thank the reviewer for this comment. Accordingly, we have changed the spelling (Line 83).

(Lines 96-110) The authors clearly have hypotheses and predictions

concerning both digit ratios and torque distributions in the different

locomotor groups. They should state these hypotheses here, rather than

frame the research in terms of questions.

Response: Thank you for providing these insights. We agree with the reviewer and have incorporated this suggestion into the Introduction (Lines 96-119).

(Methods, generally) There are many separate metrics and variables

considered in this study. A table of all abbreviations would greatly

help the reader remember what symbols are used for what metric or

variable. Also, where possible, I’d encourage the authors to use

variable names or abbreviations that are meaningful to the reader,

rather than arbitrary letters or numbers. For instance, instead of a, b,

c, and d for metacarpal and proximal, intermediate, and distal

phalangeal lengths, why not l[mc], l[pp], l[ip], and l[dp]? Similarly,

instead of tau[1], tau[2], etc for the holding torques at the joints,

why not tau[MCP] etc? Such a convention would relieve the reader from

having to remember what each symbol stands for.

Response: We thank the reviewer for this comment. In accordance with the reviewer’s comment, we have changed the symbols.

(Table 1) I don’t feel it’s appropriate to group humans with other

terrestrial primates, given that humans don’t use their hands for

locomotion. The authors seem to draw a similar conclusion much later in

the paper (lines 652-655).

Response: Thank you for providing the insight about classification of primates. We agree that humans should not be grouped with other terrestrial primates. We have further classified humans and other terrestrial primates as terrestrial bipedal and terrestrial quadrupedal primates, respectively, in Table 1. Similarly, we have changed the description of the terrestrial primates into terrestrial quadrupedal primates in the manuscript and Fig. 5

(Line328) In their review for force platform and kinematic analysis,

Biewener and Full (1992) note that calibration curves for force

transducers should typically be ≥ 0.99. I recommend fitting a quadratic

regression curve to these calibrations to improve fit and force

calculation.

Response: We thank the reviewer for this comment. The calibration curve has been precisely acquired for the range from 30g to 1,150g again. A determinant coefficient of the calibration curve was more than 0.99 using a power function as a fitting curve. The values of the torque in Fig. 8 have been recalculated, and the regression curves have also been changed.

(Line 337) hamadryas NOT hamadryases

Response: Thank you. We have corrected the spelling.

(Statistics, generally) Given that the authors are comparing 25 species,

some correction should be made for phylogenetic relatedness. Species

cannot be considered independent data points and doing so can conflate

findings that reflect phylogeny with those thought to reflect locomotion

(i.e., all terrestrial primates are catarrhines). See Barr and Scott (

2014) for a review of phylogenetic correction in the context of DFA.

Response: Thank you for providing these insights. We agree that both the phylogenetic traits and the adaptational changes are merged in the morphology of the body structures. “Adaptation (Line 683)” and “rather than phylogeny (Line 482)” have been deleted because one of them does not exclusively affect the morphology. However, we aimed to reveal the relationship between the prehensile or locomotor function and the hand structure by using the cross- or triple-ratio as a mechanical parameter of the intrinsic digital proportion in this study. In this case, regardless of whether the intrinsic digital proportion has resulted from phylogenetic signals or adaptation, we examined the relationship between the hand function and the hand structure itself. Therefore, we used CDA, but not phylogenetic generalized least square. We verified the classification result using FMA in this study.

(Figure 3) Use species abbreviations instead of arbitrary letter

abbreviations for species (i.e., Ss for Saimiri sciureus, Pp for Pongo

pygmaeus, etc).

Response: Thank you for your suggestion. We have corrected as suggested.

(Table 2) The Percent of variance and Cumulative %rows show the same

information (i.e., both are cumulative).

Response: Thank you for your suggestion. We have corrected Table 2 as suggested.

(Line 457) Scheffe’s NOT Sheffe’s

Response: Thank you for your suggestion. We have corrected as suggested.

(Figure 8) Please label the axes for each panel figure.

Response: Thank you for your suggestion. We have added the axes labels as suggested.

(Discussion, generally) Following my above recommendation to end the

introduction section with hypotheses and predictions, I recommend

returning to hypotheses and predictions in the discussion and

specifically reviewing if these predictions were met.

Response: We agree with the reviewer and have returned to the hypotheses and predictions as described in the Introduction and have reviewed if these predictions were met in the Conclusion (Lines 784-805).

(Line 705) Patel and Wunderlich should be cited in the requisite format

(i.e., numerically)

Response: Thank you for your suggestion. We have corrected as suggested (Line 756).

(Data availability) Raw data have not been made available, as per PLoS’s

policy.

Response: We thank the reviewer for this comment. Raw data have been available at the following web site.

http://hdl.handle.net/10422/00012617

---

## [Decision Letter · Decision Letter 1]

15 Apr 2020

Correlation between musculoskeletal structure of the hand and primate locomotion: Morphometric and mechanical analysis in prehension using the cross- and triple-ratios

PONE-D-19-21332R1

Dear Dr. Udagawa,

We are pleased to inform you that your manuscript has been judged scientifically suitable for publication and will be formally accepted for publication once it complies with all outstanding technical requirements.

With kind regards,

Xudong Zhang

Academic Editor

PLOS ONE

Additional Editor Comments (optional):

Reviewers' comments:

Reviewer's Responses to Questions

**Comments to the Author**

1. If the authors have adequately addressed your comments raised in a previous round of review and you feel that this manuscript is now acceptable for publication, you may indicate that here to bypass the “Comments to the Author” section, enter your conflict of interest statement in the “Confidential to Editor” section, and submit your "Accept" recommendation.

Reviewer #1: (No Response)

Reviewer #2: All comments have been addressed

2. Is the manuscript technically sound, and do the data support the conclusions?

Reviewer #1: Yes

Reviewer #2: Yes

3. Has the statistical analysis been performed appropriately and rigorously? 

Reviewer #1: Yes

Reviewer #2: Yes

4. Have the authors made all data underlying the findings in their manuscript fully available?

Reviewer #1: Yes

Reviewer #2: Yes

5. Is the manuscript presented in an intelligible fashion and written in standard English?

Reviewer #1: Yes

Reviewer #2: Yes

6. Review Comments to the Author

Reviewer #1: This remains a very challenging paper to read. I still think the predictions and hypotheses could be spelled out somewhat more clearly than they are and then linked point-by-point to the findings. I suspect that this would make the paper more useful and widely read/cited, which would be to the benefit of the authors. However, I see nothing technically wrong with the paper and no reason to delay this moving forward.

One very minor point: the authors continue using the term "brachiating" in an imprecise and non-standard fashion throughout the paper. They corrected this in places, but its inconsistent.

Reviewer #2: The authors have done a thorough job responding to my critiques and those of the other reviewer. I have no further suggestions and feel the manuscript is ready for publication.

7. PLOS authors have the option to publish the peer review history of their article (what does this mean?). If published, this will include your full peer review and any attached files.

Reviewer #1: No

Reviewer #2: Yes: Jesse W. Young

---

## [Editor Report · Acceptance letter]

22 Apr 2020

PONE-D-19-21332R1 

Correlation between musculoskeletal structure of the hand and primate locomotion: Morphometric and mechanical analysis in prehension using the cross- and triple-ratios 

Dear Dr. Udagawa:

I am pleased to inform you that your manuscript has been deemed suitable for publication in PLOS ONE. Congratulations! Your manuscript is now with our production department. 

With kind regards,

on behalf of

Dr. Xudong Zhang 

Academic Editor

PLOS ONE